

Determining the Plio-Quaternary uplift of the southern French massif-Central; a new insights for in-
traplate orogen dynamics.
Oswald Malcles[1], Philippe Vernant[1], Jean Chéry[1], Pierre Camps[1], Gaël Cazes[2,3], Jean-
François Ritz[1], David Fink[3].
[1] Geosciences Montpellier, CNRS-University of Montpellier, Montpellier, France
[2] SEES, University of Wollongong, Wollongong, Australia
[3] Australian Nuclear Science and Technology Organisation, Lucas Heights, Australia
*Correspondence to*: Oswald Malcles (oswald.malcles@umontpellier.fr)
**Abstract.**
The evolution of intra-plate orogens is still poorly understood. Yet, this is of major importance for understand-
ing the Earth and plate dynamic, as well as the link between surface and deep geodynamic processes. The
French Massif Central is an intraplate orogen with a mean elevation of 1000m, with the highest peak elevations
ranging from 1500m to 1885m. However, active deformation of the region is still debated due to scarce evi-
dence either from geomorphological or geophysical (i.e. geodesy and seismology) data. Because the Cévennes
margin allows the use of karst sediments geochronology and morphometrical analysis, we study the vertical dis-
placements of that region: the southern part of the French Massif-Central. Geochronology and morphometrical
results, helped with lithospheric-scale numerical modelling, allow, then, a better understanding of this intraplate-
orogen evolution and dynamic.
Using the ability of the karst to durably record morphological evolution, we first quantify the incision
rates. We then investigate tilting of geomorphological benchmarks by means of a high-resolution DEM. We fi-
nally use the newly quantified incision rates to constrain numerical models and compare the results with the ge-
omorphometric study.
We show that absolute burial age ($^{10}$Be/$^{26}$Al on quartz cobbles) and the paleomagnetic analysis of karstic clay
deposits for multiple cave system over a large elevation range correlate consistently. This correlation indicates a
regional incision rate of 83.4 $^{+17.3}/_{-5.4}$ m.Ma$^{-1}$ during the last ca 4 Myrs (Plio-Quaternary). Moreover, we point out
through the analysis of 55 morphological benchmarks that the studied region has undergone a regional south-
ward tilting. This tilting is expected as being due to a differential vertical motion between the north and southern
part of the studied area.
Numerical models show that erosion-induced isostatic rebound can explain up to two-thirds of the regional up-
lift deduced from dating technics and are consistent with the southward tilting obtain from morphological analy-
sis. We presume that the remaining part is related to dynamic topography or thermal isostasy due to the Massif
Central plio-quaternary magmatism.

## 1 Introduction and Tectonic Setting

Since the past few decades, plate-boundary dynamics is to a first order, well understood. Such is not the case for
intraplate regions, where short-term ($10^3$-$10^5$ yrs.) strain rates are low and the underlying dynamical processes





are still in debate (e.g. Calais et al., 2010; Vernant et al., 2013; Calais et al., 2016; Tarayoun et al., 2017). On ge-
ological time-scales, transient phenomenon that are classically used to explain intraplate deformations (as seen
through the seismic activity) can't be a satisfactory explanation though, this then raises the question of the origin
of the high finite deformations observed in many parts of the world as for instance the Ural mountains in Russia,
the Blue Mountains in Australia or the French Massif Central.
In this study we focus on the Cevennes Mountains and the Grands Causses regions that form the southern part
of the French Massif Central, located in the southwestern Eurasian plate (fig.1). The region is characterized by a
mean elevation of 1000 m with summits higher than 1500 m. Such topography is the result of recent, active up-
lift and as the Cevennes mountains experiences an exceptionally high mean annual rainfall (the highest peak,
Mount Aigoual, records the highest mean annual rainfall in France of 4015 mm) it raises the question of a possi-
ble link between erosion and uplift as previously proposed for the Alps (Champagnac et al., 2007; Vernant et al.,
2013; Nocquet et al., 2016). This region currently undergoes a small but discernible deformation, but no signifi-
cant quantification can be deduced due to the scarcity in seismicity (Manchuel et al., 2018).  In addition, GPS
velocities are below the uncertainty threshold of GPS analyses (Nocquet et Calais, 2003; Nguyen et al., 2016).
South and West of the crystalline Cevennes mountains, prominent limestone plateaus, named Grands Causses,
rise to 1000m and are dissected by several canyons. The initiation of incision, its duration and the geomorphic
processes leading to the present-day landscape remain poorly constrained. A better understanding of the pro-
cesses responsible for this singular landscape would bring valuable information on intraplate dynamics, espe-
cially where large relief exists.
The oldest formations in the area were formed during the Variscan orogeny (late Palaeozoic, ~300 Myrs ago;
Brichau et al., 2007) and constitute the crystalline basement of the Cevennes. Between 200 and 40 Myrs ago
(Mesozoic and lower Cenozoic), the region was mainly covered by the sea ensuring the development of an im-
portant detrital and carbonate sedimentary cover, which can be more than 9 km thick in some locations (Sanchis
and Séranne, 2000; Barbarand et al., 2001). During the Mesozoic era, an episode of regional erosion and alter-
ation (called the Durancian isthmus) is proposed as being at the origin of the flat, highly elevated surface that
persists today across the landscape (Bruxelles, 2001; Husson, 2014).
The area is also affected by the major NE-SW trending Cevennes fault system. During the Pyrenean orogeny, 85
to 25 Ma (Tricart, 1984; Sibuet et al., 2004), several faults and folds affected the geological formations south of
the Cevennes fault, while very few deformations occurred further north within the Cévennes and Grand Causses
areas (Arthaud and Laurent, 1995). Eventually, the Oligocene extension (~30 Myrs ago) led to the counterclock-
wise rotation of the Corso-Sardinian block and the opening of the Gulf of Lion, re-activating some of the older
compressive structures as normal faults. The main drainage divide between the Atlantic Ocean and the Mediter-
ranean Sea is located in our study area and is inherited from this extensional episode (Séranne et al., 1995; San-
chis et al., 2000).
These two tectonic episodes (Pyrenean compression and Oligocene extension) are the main geodynamic pro-
cesses that shaped the large-scale structural morphology of the region. Afterwards during the Plio-Quaternary
period, only intense volcanic activity has affected the region, from the Massif Central to the Mediterranean
shoreline. This activity is characterised by several volcanic events that are well constrained in age (Dautria et
al., 2010). The last eruption occurred in the Chaîne des Puys during the Holocene (i.e. the past 10 kyrs (Nehlig





et al., 2003; Miallier et al., 2004). Some authors proposed that this activity is related to a hotspot underneath the
Massif Central (Granet et al., 1995; Baruol and Granet, 2002) leading to an observed positive heat-flow anomaly
and a possible regional plio-Quaternary uplift.

78          Despite this well described overall geological evolution the onset of active incision that has shaped the

deep valleys and canyons (e. g. Tarn or Vis river, Fig 1) across the plateaus, and the mechanisms that controlled
this incision are still in debate. One hypothesis proposes that canyon formation was driven by the Messinian
salinity crisis with a drop of more than 1000m in Mediterranean Sea level. This, however, would then not ex-
plain the fact that the Atlantic watersheds show similar incision. Other studies suggested that the incision is con-
trolled by the collapse of cave galleries that lead to fast canyon formation mostly during the late Quaternary,
thus placing the onset of canyon formation only a few hundreds of thousands of years ago (Corbel, 1954). In
contrast, it has also been proposed more recently (based on relative dating techniques and sedimentary evidence)
that incision during the Quaternary was negligible (i.e. less than a few tens of meters), and that the regional mor-
phological structures seen today occurred around 10 Myrs ago (Séranne et al., 2002; Camus, 2003).

88          In this paper, we provide new quantitative constraints on both the timing of incision and the rate of riv-

er down-cutting in the central part of the Cévennes and of the Grands Causses that has resulted in the large relief
between plateau and channel bed.  We employ two methods, cosmogenic $^{10}Be/^{26}Al$ burial dating quartz cobbles
that have been transported by rivers and paleomagnetic analyses along vertical profiles of endokarstic clay both
of which have been deposited in multiple cave systems at the time cave entry was at river channel elevation. In
parallel, by analysing a high-resolution DEM (5m), we show that the region is affected by a regional tilt. Our re-
sults allow to quantify the role of the Plio-Quaternary incision on the Cévennes landscape evolution and to con-
strain numerical modelling from which we derive the regional uplift rates and a tilt of geomorphological mark-
ers.

97          One important point of this study is the integration of multi-disciplinary approaches in order to con-

strain intraplate deformation. Such an approach is necessary to bring new insights into the lithosphere behaviour
of slow dynamic regions. If the uplift is easily recognisable in the landscape (1000 m high plateaus), quantifying
its timing and evolution rates is harder and can't be performed by classical technics (e.g. GPS). This is why we
aim to quantify the incision rate over the longest possible period thanks to the karstic immunity. Dealing with
long-term incision rates (up to 5 Myrs) should permit to smooth possible climatic-driven incision rate variations
(with time-span of several kyrs).
If incision is initiated by uplift centred on the North of the area where elevations are maximum, it will lead to
tilting of fossilised topographic markers as strath terraces.  Our method of analyses provides an opportunity to
select between three possible explanations for the current terrain morphology. The first is based on old uplift
and old incision (Fig. 2.A). In this case, apparent incision rates would be very low. For instance, if incision com-
menced 10 Myrs ago (Serrane et al., 2002), we would find surface tilting but cosmogenic burial dating with
$^{10}Be/^{26}Al$ which cannot discern ages older than ~ 5Ma due to excessive decay of 26Al, would not be possible.
The second possibility (Fig. 2.B) is that the uplift is old, and incision consequently follows but with a time lag.
Here the incision rate would be rather fast but no tilting is expected for the river-related markers because no dif-
ferential uplift occurs after their formation. Finally, the third possibility (Fig 2.C) is that uplift and incision are



concurrent and recent (i.e. within the time scale of cosmogenic burial dating) and thus we would expect burial
ages < 5 Myrs relatively high incision rates, and  tilting of morphological markers.  These different proposals for
the temporal evolution of the region will then be compared using numerical modelling.
**2. Determining the incision rates in the Cévennes and the Grand Causses Region**
**2.1. Principles and methods**
**2.1.1. Karst model**
No evidence of important aggradation events has been reported in the literature for the studied area. Therefore
we base our analysis on a per descensum infill model of the karst networks whereby sediments are transported
and then deposited within cave galleries close to base level. When cave-systems and entry passages are near the
contemporaneous river channel elevation (including higher levels during floods), the deposition into caves of
sediments, from clay to cobbles occurs, especially during flood events. With subsequent river incision into
bedrock creating a relative base level drop (due to uplift or sea-level variations). The galleries associated with
the former base-level are now elevated above the new river course and become disconnected from further depo-
sition. Hence fossilised and trapped sediments throughout the cave network represent the cumulative result of
incision. In this commonly used model (Granger et al., 1997; Audra et al., 2001; Stock et al., 2005; Harmand et
al., 2017), the higher the gallery elevation (relative to the present-day base level) the older the deposits in that
gallery. As a result, the objective here is to quantify a relative lowering of the base level in the karst systems,
with the sediments closest to the base level being the youngest deposits, and note that we do not date the cave
network creation which may very well pre-date river sediment deposition.
Within individual canyons, successions of gallery networks across the full elevation range from plateau top to
modern river channel, were not always present and often sampling could not be conducted in a single vertical
transect. Thus we make the assumption of lateral altitudinal continuity i.e. that within a watershed, which may
contain a number of canyons, the sediments found in galleries at the same elevation were deposited at the same
time. Inside one gallery, we use the classical principle of stratigraphy sequence (i.e. the older deposits are below
the younger ones).
**2.1.2. Burial ages**
Burial dating using Terrestrial cosmogenic nuclides (TCN) are nowadays a common tool to quantify incision
rates in karstic environment (Granger et Muzikar, 2001; Stock et al., 2005; Moccochain., 2007; Tassy et al.,
2013; Granger et al., 2015; Calvet et al., 2015; Genti, 2015; Olivetti et al., 2016; Harmand et al., 2017; Rovey II
et al., 2017; Rolland et al., 2017; Sartégou, 2017; Sartégou et al., 2018).  This method relies on the differential
decay of TCN in detrital rocks that were previously exposed to cosmic radiation before being trapped in the
cave system. With this in mind, the $^{10}Be$ and $^{26}Al$ nuclide pair is classically used as (i) both nuclides are pro-
duced in the same mineral (i.e. quartz), (ii) their relative production ratio is relatively well constrained (we use
here a  standard $^{26}Al/^{10}Be$ pre-burial ratio of 6.75, see Balco et al., 2008) and (iii) their respective half-lives





(about 1.39 Myr and 0.70 Myr for $^{10}$Be and $^{26}$Al, respectively) are well suited to karstic and landscape evolution
study, with a useful time range of ~100 ky to ~5 Myr.
To quantify the incision rate of the limestone plateau of the Cevennes area, we analysed quartz cobbles infilling
from four caves of the Rieutord canyon (Fig. 1), this canyon is well suited for such study because horizontal
cave levels are tiers over 200 m above the current river-level and are directly connected to the canyon, leading
to a straight relationship between river elevation and the four cave infilling that we have sampled (Cuillère cave,
Route cave, Camp-de-Guerre cave and Dugou cave). Furthermore, cobbles source is well known and identified:
the upstream part of the Rieutord river, some tens of kilometres northward, providing a uniform sediment origin.
All samples (Example Fig. 3) were collected far enough away (>20m) from the cave entrance and deep enough
below the surface (>30m) to avoid secondary in-situ cosmogenic production of $^{10}$Be and $^{26}$Al in the buried sedi-
ments.
The quartz cobbles were first crushed and purified for their quartz fraction by means of sequential acid attack
with Aqua-Regia (HNO$_3$ +3HCl) and diluted Hydrofluoric acid (HF). Samples were then prepared according to
ANSTO's protocol (see Child et al. 2000) and ~300μg of a $^9$Be carrier solution was added to the purified quartz
powder before total dissolution. AMS measurements were performed on the 6MV SIRIUS AMS instrument at
ANSTO and results were normalised to KN-5-2 (for Be, see Nishiizumi et al., 2007) and KN-4-2 (for Al) stan-
dards. Uncertainties for the final $^{10}$Be and $^{26}$Al concentrations include AMS statistics, 2% (Be) and 3% (Al) stan-
dard reproducibility, 1% uncertainty in the Be carrier solution concentration and 4% uncertainty in the natural
Al measurement made by ICP-OES, in quadrature. Sample-specific details and results are found in table 1.
For the four caves, we observed a good relationship between burial ages and incision, except for the Cam-
p-de-Guerre cave (CDG) site, the higher the cave is, the older the burial ages are. Burial ages for the Cuillère
cave, Dugou cave, Camp-de-Guerre cave and Route cave are 2.16 ± 0.154, 0.95 ± 0.137, 0.63 ± 0.097 and 0.21
± 0.1 Myrs respectively.
**2.1.3. Paleomagnetic analysis**
In parallel with burial dating, we collected 141 clay-infilling samples into two main cave systems: the
*Grotte-Exsurgence du Garrel* and the *Aven de la Leicasse* (Fig. 1). These two sites allowed us collecting sam-
ples along a more continuous range of elevations than the one provided by the Rieutord samples and also al-
lowed extending the spatial coverage to the Southern Grands Causses region. Thanks to the geometry of these
two cave systems, we sampled a 400m downward base level variation. The sampling was done by means of
Plexiglas cubes with a 2 cm edge length (Fig. 4) used as a pastry cutter. We weren't able to analyse clay samples
from Rieutord canyon because no reliable clay infilling was found in the Rieutord caves.
Demagnetisation was performed with an applied alternative field up to 150mT using a 2G-760 cryogenic mag-
netometer, equipped with the 2G-600 degausser system controller. Before this analysis, each sample remained at
least 48h in a null magnetic field, preventing a possible low coercivity viscosity overprinting the detrital rema-
nent magnetisation (DRM) (Hill, 1999; Stock et al., 2005; Hajna et al., 2010). If the hypothesis of instantaneous
locked in DRM seems reasonable compared with the studied time span, it is important to keep in mind that the
details of DRM processes (as for instance the locked in time) is not well understood (Tauxe et al., 2006; Spassov





et Valet, 2012) and could possibly lead to small variations (few percents) in the following computed incision
rates.
Because fine clay particles are expected being easily reworked in the cave, careful attention was paid to the site
selection and current active galleries were avoided. Clays deposits had to show well laminated and horizontal
layering in order to prevent analysis of in-situ produced clays (from decalcification) or downward drainage by
an underneath diversion gallery that could strongly affect the obtained inclination (and also the declination to a
minor extent). Note that for paleo-polarities study alone, small inclination or declination variations won't result
in false polarities

## 2.2 Quantifying the average incision rates

### 2.2.1. Rieutord incision rate from burial ages

The relationship between burial ages and incision is shown in Figure 5. Except for the Camp-de-Guerre
cave (CDG) site, the higher the cave is, the older the burial ages are. This is consistent with the supposed cave
evolution and first-order constant incision of the Rieutord canyon. CDG age has to be considered with caution.
The CDG cave entrance located in a usually dry thalweg can act as a sinkhole or an overflowing spring depend-
ing on the intensity of the rainfall.  The sample was collected in a gallery showing evidence of active flooding
~10 m above the Rieutord riverbed, therefore the older than expected age, given the elevation of the cave, is
probably due to cobbles that came from upper galleries during flood events. Forcing the linear regression to go
through the origin, leads to an incision rate of $82.8 \pm 34.9$ m.Ma$^{-1}$. These results show that at least half of the
300 m deep Rieutord Canyon is a Quaternary incision. Extrapolating the obtained rate yields an age of $4.4 \pm 1.9$
Ma for the beginning of the canyon incision, which suggests that the current landscape has been shaped during
the Plio-Quarternary period. To extend our spatial coverage and bring stronger confidence into our results, we
combine Rieutord burial ages with paleomagnetic data from watersheds located on the other side of the Herault
watershed.

### 2.2.2. South Grands Causses incision rate from paleomagnetic data

Of the two cave systems, the lowest sample elevation above sea level (a.s.l.) is in the Garrel (ca 190 m) and the
highest in the Leicasse (ca 580 m a.s.l.). Given the very marginal difference in elevation between the local base
levels from these two caves, we assume that they have the same local base level reference. In the Leicasse cave
system, we sampled 8 sites for a total of 61 samples. Their elevations are located between ca 200 m and ca 400
m above base level (a.b.l.), defined as the elevation of the Buèges river spring at 170 m above sea level.
From 5 sites, the 80 Garrel samples encompass elevations range between 20 m and 80 m a.b.l. defined by the
Garrel spring at 180 m a.s.l.
Because one site is a vertical profile of samples and can count between 3 to 15 samples (fig. 4), the figure 6 rep-
resents the magnetic polarities by individual sites, in respect with their elevation a.b.l. If all the samples of one
site have the same polarity, the site is granted with the same polarity. If not, that is to say if the site displays nor-
mal and reverse polarities, we consider it as a transitional site.
First, we note a good agreement between samples located at the same elevation, preventing from a possible par-



tial endokarstic reworking. Second, the different elevations of the galleries where we collected the samples al-
low us to propose that the Leicasse and the Garrel deposits encompass at least three and one polarity chrons, re-
spectively. Furthermore, Les Gours sur Pattes (LGP) sampling site record a reversal signal, the lower samples
being "reverse", the upper ones "normal" and the sample in between show a transitional signal (Fig. 7). This
specific site provides strong constraints on the age of the sediment emplacement in the magnetostratigraphic
timescale (Fig. 6).
Although poorly constrained since it relies on a single sample with reverse polarity at (90 m a.b.l.), the eleva-
tion/polarity results for the Garrel agrees with U-Th ages younger than 90 kyrs obtained for two speleothems
that cover our sampled clays in the Garrel at ca 40 m a.b.l. (Camus, 2003) (Fig. 6). Since no reversed polarities
have been found beneath the speleothems despite 72 collected samples, we assume that the emplacement of
these clays deposits occurred during the most recent normal period and are therefore younger than 0.78 Ma. The
transition between the highest normal sample and the reversed one is located somewhere between 78 m and 93
m a.b.l. suggesting a base level lowering rate of $109.6 \pm 9$ m.Ma$^{-1}$.
Unfortunately our sampling resolution prevent us from studying possible variations of the incision rate through
time. Given our results for the Rieutord samples we assume that the incision rate can be considered to the first
order as linear through time. Therefore, we computed theoretical age models for every clay sample, from both
the Garrel and the Leicasse cave systems, as a function of various incision rates ranging from 0 to 200 m.Ma$^{-1}$
with a 1 m.Ma$^{-1}$ step. Then, from the magnetostratigraphic timescale, we extracted the theoretical polarity for
each sample and computed a correlation factor based on the consistency between the observed polarities and the
modelled ones.  We obtained 10 possible incision rates with the same best correlation factor (Fig. 8) spanning
from 43 to 111 m.Ma$^{-1}$ (mean of $87.2 \pm 23.8$ m.Ma$^{-1}$). Taking into account the transitional signal of the LGP site
in the Leicasse cave yields a linear incision rate of $83.4\ ^{+17.3}/_{-5.4}$ m.Ma$^{-1}$. Proposed uncertainties are based on pre-
vious and next transition-related estimated incision rate.
Using a similar approach for the Rieutord crystalline samples, we determined a linear incision rate of $85 \pm 11$
m.Ma$^{-1}$ (Fig 8). Those two results, based on independent computations, suggest the same first-order incision rate
for the last 4 Ma of $84.2\ ^{+20.5}/_{-12.3}$ m.Ma$^{-1}$. Given that the Rieutord, Garrel and Buèges rivers are all tributaries of
the Hérault river, we propose that this rate represents the incision rate for the Hérault river watershed, inducing
approximately 300-350 m of finite incision over the Plio-Quaternary period.
If the landscape is at first order in an equilibrium state, that is to say, if we preclude our incision rates being a re-
gressive erosional signal, the incision needs to be balanced by an equivalent amount of uplift. If the uplift rate is
roughly correlated to the regional topography, lowest uplift rates would be expected in the south of our sampling
sites inducing regional tilting of morphological benchmarks. In the next part, we search for such evidences that
would suggest differential uplift.
2.3 **Geomorphometrical evidence**
According to the Massif-Central centered uplift hypothesis, morphological markers such as strath terraces, flu-
vio-karstic surfaces or abandoned meanders should display a southward tilting due to differential uplift between
the northern and the southern part of the region, with the expected following signals:
- The dipping direction of the tilted markers should be parallel to the main gradient of the topography, i.e. be-





tween 150°E and 180°E for our studied region. This expectation is the most important one, regarding uncertain-
ties on the uplift rate and lithospheric elastic parameters.
- A latitudinal tilting trend, i.e. an increase of the tilt angle along the topography gradient. Indeed, null or small
tilts are expected near the shoreline and within the maximum uplift area of the Cevennes/Massif Central, while
the maximum tilt is expected at a mid-distance between these two regions, i.e. about 50 km inland from the
shoreline.
- A positive altitudinal tilting trend (an increase in dip angle with altitude). This trend would be representative
of the accumulation of finite tilt. However, it supposes a linear relationship between the altitude and the age of
the marker formation. If at first order, this straightforward hypothesis seems reasonable for river-controlled
markers (e.g. strath terraces), other surfaces are hardly expected to follow such an easy relationship.

268        To investigate these different signals, we used the morphological markers available for the study area

(Fig. 9). We used a 5 m resolution DEM analysis to identify the markers corresponding to surfaces with slope <
2°. This cut-off slope angle prevents to identify surface related to local deformation such as for example land-
slide or sinkhole. The local river slope is on the order of 0.1° so the 2° cut-off angle is far from precluding to
identify tilted markers. We also us a criterion based on an altitudinal range for a surface. This altitudinal span is
set individually for each surface based on elevation, slope and curves map analysis, and encompass from few
meters to tens of meters depending on the size of the marker. We checked 80% of the identified surfaces in the
field in order to avoid misinterpretation. The dip direction and angle of the surface in computed in a two steps
approach. First, we compute a plan using extracted points from the DEM inside the delimited surface. Second,
based on this plan we remove the DEM points with residuals 3 times larger than the standard error and compute
more accurate plan parameters. This outlier suppression removes any inaccurate DEM points and correct for in-
accurate surface delimitation (e.g. integration of a part of the edge of a strath terrace).
Because no obvious initially horizontal markers are known, we propose to correct the marker current slope by
the initial one to quantify the tilt since the marker emplacement. To do so we follow the method used by Cham-
pagnac et al. (2008) for the Forealps. We identify the drain related to the marker formation and compute its cur-
rent local slope and direction. This method assumes that landscapes are at the equilibrium state and that the river
slope remained constant since the marker formation. This assumption seems reasonable given the major river
profiles and because most of the markers used are far from the watershed high altitude areas precluding a reces-
sive erosional signal. Finally, we removed the local river plan from the DEM extracted surface.
Following this methodology, we obtained 61 surfaces. We then applied three quality criterions to ensure the ro-
bustness of our results: 1) The minimal surface considered is 2500 m² based on a comparison between the 5m
resolution DEM and a RTK GPS survey over 3 strath terraces (Hérault river); 2) Final plans with dip angles
larger than 2° are removed; 3) The residuals for each geomorphological marker must be randomly distributed
without marker edge signal, or clear secondary structuration. Only 38 markers meet those 3 quality criterions.
The results show a mean tilt angle of 0.61 ± 0.41 ° with an azimuth of N150 ± 40°E (Fig. 10).

## 3 Numerical modelling

Both geomorphological and geochronological evidence suggest a Plio-Quaternary uplift of the Cevennes area.



The origin of such uplift could be associated with several processes: erosion-induced isostatic rebound, dynamic
topography due to mantle convection, thermal isostasy, residual flexural response due to the Gulf of Lion forma-
tion, etc. For the Alps and Pyrenees mountains, isostatic adjustment due to erosion and glacial unloading has
been recently quantified (Champagnac et al., 2007, Vernant et al., 2013; Genti et al, 2016, Chery et al. 2016).
Because the erosion rates measured in the Cevennes are similar to those of the Eastern Pyrenees (Calvet et al.,
2015, Sartégou et al., 2018a), we investigate by numerical modelling how an erosion-induced isostatic rebound
could impact the southern Massif Central morphology and deformation.
We define a representative cross-section parallel to the main topographic gradient (i.e. NNW-SSE) and close to
the field investigation areas (Figure 11). We study the lithospheric elastic response to erosion with the 2D finite
element model ADELI (Hassani et Chery, 1996; Chéry et al. 2016). The model is composed of a plate account-
ing for the elasticity of both crust and uppermost mantle. Although the lithosphere rigidity of the European plate
in southern Massif central is not precisely known, vertical gradient temperatures provided by borehole measure-
ments are consistent with heat flow values ranging from 60 to 70 mW.m$^2$ (Lucazeau et Vasseur, 1989). There-
fore, we investigate plate thickness ranging from 10 to 50 km as done by Stewart et Watts (1997) for studying
the vertical motion of the alpine forelands. We choose values for Young's and Poisson parameters of respective-
ly 10$^{11}$ Pa and 0.25, both commonly used values for lithospheric modelling (e.g. Kooi et Cloething, 1992;
Champagnac et al. 2007, Chéry et al., 2001). This leads to long-term rigidity of the lithosphere model ranging
from 10$^{21}$ to 10$^{25}$ N.m. Since the effect of mantle viscosity on elastic rebound is assumed to be negligible at the
time scale of our models (1 to 2 Myrs), we neglect the visco-elastic behaviour of the mantle. Therefore, the base
of the model is supported by an hydrostatic pressure boundary condition balancing the weight of the lithosphere
(Fig. 11). Horizontal displacements on vertical sides are set to zero since geodetic measurements show no sig-
nificant displacements (Nocquet et Calais, 2003; Nguyen et al., 2016). The main parameters controlling our
model are the erosion (or sedimentation) triggering isostatic rebound and the elastic thickness. The erosion pro-
file (Fig. 11) is based on topography, our newly proposed incision rate and other studies (Olivetti et al., 2016 for
onshore denudation and Lofi et al., 2003; Leroux et al., 2014 for offshore sedimentation). The flexural rigidity
controls the intensity and wavelength of the flexural response and ranges from 10$^{21}$ to 10$^{25}$ N.m. It can be ex-
pressed as a variation in elastic thickness (Te) ranging from 4.4 to 96 km (Fig. 12). We also test a possible Te
variation between inland and offshore areas. For the following discussion, we use an elastic thickness of 15km
corresponding to a value of D of 3.75 x10$^{23}$ N.m$^{-1}$. In this case, the inland and offshore parts are largely decou-
pled and the large sedimentation rate in the Gulf of Lion does not induce a flexural response on the Cévennes
and Grands Causses areas. With a maximum erosion rate of 80 m.Ma$^{-1}$ (Fig 11), the models display uplift rates
of 50 m.Ma$^{-1}$ over more than 100 km. As previously explained, the finite incision is permitted by an equal
amount of uplift considering that the incision is not due to regressive erosion. If all tested models show uplift,
the modelled amplitudes are smaller than the expected ones. To obtain the same uplift rate than the incision
rates, the applied erosion rate over the model must be increased. However, we assume that the landscape is at
equilibrium, so, if the erosion rate is increased, it will be higher than the incision rate leading to the decay of re-
lief over the area. No evidence of such evolution is found over the region and, if further studies need to be done
to quantify the actual erosion rate, we mostly think that a second process is acting, inducing the rest of the uplift
that can't be obtained by the erosion–induced isostatic adjustment. Finally, models predict a seaward tilt of the



surface at the regional-scale (Fig. 13), in agreement with the observed tilting of morphological markers.
4**. Discussion**
We assume that the sediments collected in the karst were deposited per descensum, i.e. we do not know
if the galleries existed a long time before or were formed just before the emplacement of the sediments, but the
more elevated the sediments are, the older their deposit is. If there is no evidence of an important aggradation
episode leading to more a complex evolution as proposed for the Ardèche canyon (Moccochain et al., 2007;
Tassy et al., 2013), we point out that small aggradation or null erosion period could, however, be possible. Some
processes could explain such relative stability: e.g. variation in erosion (due to climatic fluctuation) or impact of
eustatic variations (in river profile, flexural response, etc.). Such transient variations have been shown for the
Alps (Saillard et al., 2014; Rolland et al., 2017) and are proposed as being related to climato-eustatic variations
and therefore should last 10 to 100 kyrs at most.
Based on our sampling resolution, we cannot evidence such transient periods and we must use an average base
level lowering rate in the karst, which we correlate to the incision of the main rivers. The TCN-based incision
rate derived from the Rieutord samples ($82.8 \pm 34.9$ m.Ma$^{-1}$) is consistent with the one derived from the Garrel
(U-Th ages: 85.83 m.Ma$^{-1}$ according to the sole U/Th exploitable result (Camus, 2003)) and from the Garrel-Le-
icasse combination (Paleomagnetic approach: $84.2\ ^{+20.5}/_{-12.3}$m.Ma$^{-1}$).
This mean incision rate of ca. 85 m.Ma$^{-1}$ lasting at least 4 Ma, highlights the importance of the Plio-Quaternary
period into the Cévennes and Grand Causses morphogenesis. Furthermore, the 300 to 400 m of incision pre-
cludes a relative base level controlled by a sea-level drop. Indeed, documented sea level variations are less than
100 m (Haq, 1988, Miller et al., 2005). Furthermore, the Herault river does not show any significant knickpoints
or evidence of unsteadiness in its profile as expected if the incision was due to eustatic variations. Therefore, we
propose that the incision rate of ~85 m.Ma$^{-1}$ is due to a plio-quaternary uplift of the Cévennes and Grands
Causses region.
Other river-valley processes could lead to a local apparent high incision rate as for instance major land-
slide or alluvial fan (Ouimet et al., 2008). This hypothesis of an epigenetic formation of the Rieutord is irrele-
vant because of i) none of the possible causes had been found in the Rieutord canyon and ii) the consistency of
the TCN-based incision rate and the paleomagnetic-based incision rate for two other cave-systems. Indeed, the
use of two independent approaches and three locations is a good argument in favour of the robustness of our
proposed mean 85 m.Ma$^{-1}$ incision rate. Yet, using more data, particularly burial dating colocalized with clays
samples and adding sampling sites would give a stronger statistical validation. In the Lodève basin (Point 4, fig.
1), inverted reliefs allow another independent way to quantify minimal incision rate. K/Ar and paleomagnetic
dated basaltic flows spanning from 1 to 2 Myrs old that were deposited at the bottom of the former valley
(Dautria et al., 2010) are now located at ca 150 m above the current riverbed leading to an average incision rate
of $76.5 \pm 10$ m.Myr$^{-1}$, in agreement with karst-inferred incision rates.
Furthermore, preliminary results from canyons on the other side of the Grands Causses (Tarn and Jonte) based
on in-situ terrestrial cosmogenic dating suggest similar incision rates (Sartegou et al., 2018b) and confirm a re-
gional base level lowering of the Cévennes and Grands Causses region during the Plio-Quaternary. This is con-



sistent with the similarities of landscapes and lithologies observed both on the Atlantic and Mediterranean wa-
tersheds (e.g. Tarn river).
Once the regional pattern of the Plio-Quaternary incision established for the Cévennes-Grands Causses
area, the next question is how this river downcutting is related to the regional uplift? First order equilibrium
shape and absence of major knick points in the main river profiles preclude the hypothesis of regressive erosion.
Hence, the incision rate has to be balanced to the first order by the uplift rate. No obvious evidence of active tec-
tonic is reported for the area raising the question of the processes responsible for this regional uplift. Very few
denudation rates are reported for our study area (Schaller et al., 2001; Molliex et al., 2016; Olivetti et al., 2017),
and converting canyon incision rates into denudation and erosion rates is not straightforward, especially given
the large karst developed in the area. Using a first order erosion/sedimentation profile following the main topog-
raphy gradient direction we have modelled the erosion-induced isostatic rebound. If this process could create be-
tween half and two third of the Plio-Quaternary uplift, a previously existent topography is needed to trigger ero-
sion so it cannot explain neither the onset of the canyon-carving nor the full uplift rates. Other, processes have to
be explored such as dynamic topography or thermal anomaly beneath the Massif-Central, the magmatism re-
sponsible for the important increase in volcanic activity since ~ 6 Myrs (Michon et Merle, 2001; Nehlig et al.,
2003) could play a major role, notably in the initiation of Plio-Quaternary uplift.
**5. Conclusion**
To the contrary of previous studies that focused on one cave, we have shown that combining karst buri-
al ages and paleomagnetic analysis of clay deposits in several caves over a large elevation range can bring good
constraints on incision rates. This multi-cave system approach diminishes the intrinsic limits of the two single
methods: low sampling density (and analysis cost) for the TCN ages and difficulty to set the position of paleo-
magnetic results. Our estimated paleo base level ages are Plio-Quaternary (ca. last 4 Ma) and allow to derive a
mean incision rate of 83.4 $^{+17.3}/_{-5.4}$ m.Ma$^{-1}$ for the Cevennes area.
The landscape, and especially the river profiles suggest a first-order equilibrium allowing considering
the incision rate as an uplift rate. We propose that related erosional isostatic adjustment is of major importance
for the understanding of the southern French Massif-Central landscape evolution and explain a large part of the
uplift. However, it is not the only process involved and we hypothesize that is could be especially combined
with dynamic topography related to the Massif Central magmatism. Both mechanisms imply an uplift centered
on the Massif Central and a radial tilt of the geomorphological surfaces. We have shown using a geomorpholog-
ical analysis that at least south of the Cévennes, several surfaces are tilted toward the SSE. This kind of study
had been performed before on large structures (Champagnac et al., 2007) or endokarstic markers (Granger et
Stock, 2004) but it is the first time that it is performed at such scale with small markers. Numerical modelling
yields the same pattern of SSE dipping, allowing more confidence in the geomorphometric results.
Our multi-disciplinary approach brings the first absolute dating of the Cévennes landscapes and suggests that the
present-day morphology is partly inherited from the plio-quaternary erosion-induced isostatic rebound. A strong
uplift impact is assumed to be due to magmatic-related dynamic topography that could explain another part of
the uplift as well as the onset of such uplift that has afterward been accelerated by the erosion-induced isostatic
rebound. These results enlighten the importance of surface processes into lithospheric-scale dynamic and verti-



cal deformations in intra-plate domains.
An analysis at the scale of the Massif Central is now needed before nailing down our interpretations,
but such study will more likely highlight the importance of erosion processes to explain uplift of intraplate oro-
gens, and will show that another process is needed for the Massif Central, which will most likely be dynamic to-
pography related to magmatism.

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

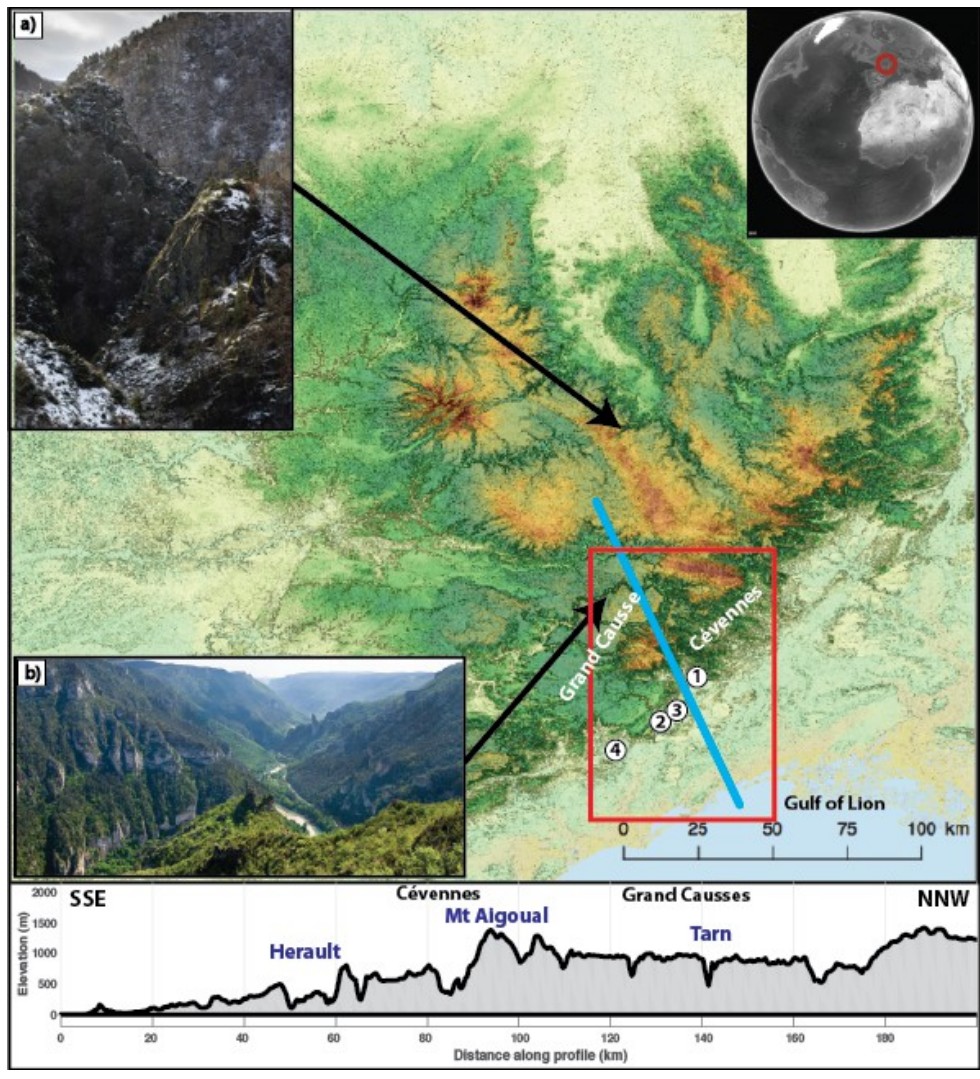

**Figure 1:** 30 m resolution DEM of the French Massif-Central and slope shadowed. Examples of finite incision typical of the French Massif-Central in a) cristalline area (Seuge Canyon) and b) limestone plateau (Tarn Canyon) Location of the restricted studied area in red box (fig. 8) and numerated site 1) is the Rieutord Canyon, 2) is the Leicasse Cave System and 3) is the Garrel Cave system and 4) is the Lodève bassin with dated basaltic flows. Bottom panel is an example of typical topographic profile used for numerical model set up.



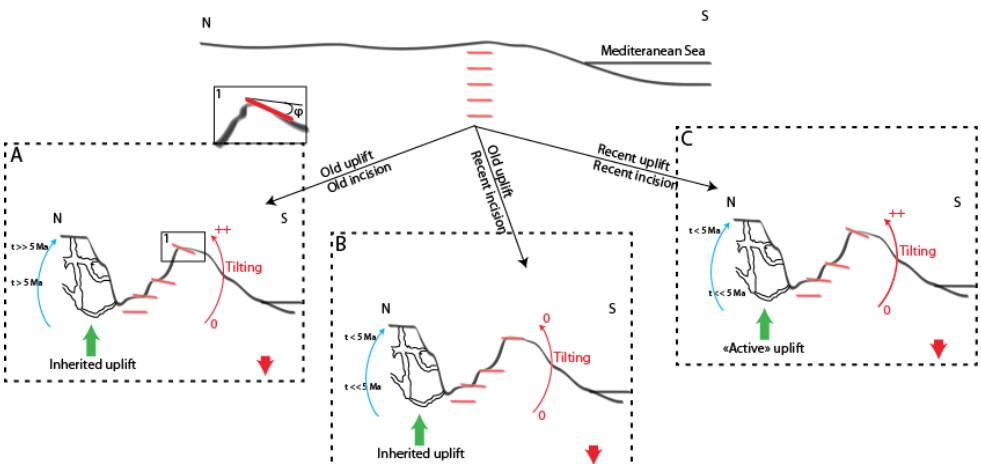

**Figure 2: conceptual models for landscape evolution. Top panel is the initial stage (prior to uplift). Each panel represent a possible scenario explaining current morphology: A) Old uplift and old incision, B) Old uplift and recent incision and C) both recent uplift and incision. Blue arrow and associated ages show expected result (or absence of) for burial dating. Red level represents morphological markers that are fossilised when reaching the surface, accumulating afterward (or not) the differential uplift by finite tilting.**

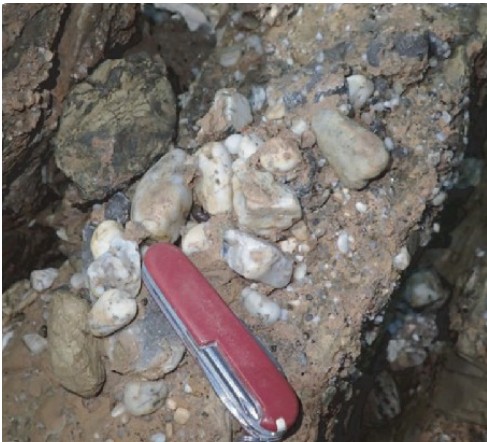

**Figure 3: Example of quartz cobbles sampled for burial dating. Location: Cuillère Cave**



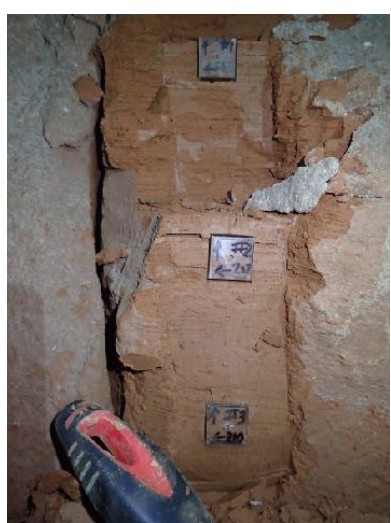

**Figure 4: Example of clay sampling for the paleomagnetic study. Location at the entrance shaft (Highest elevation of every samples (~580 m a.s.l.), Leicasse Cave system)**

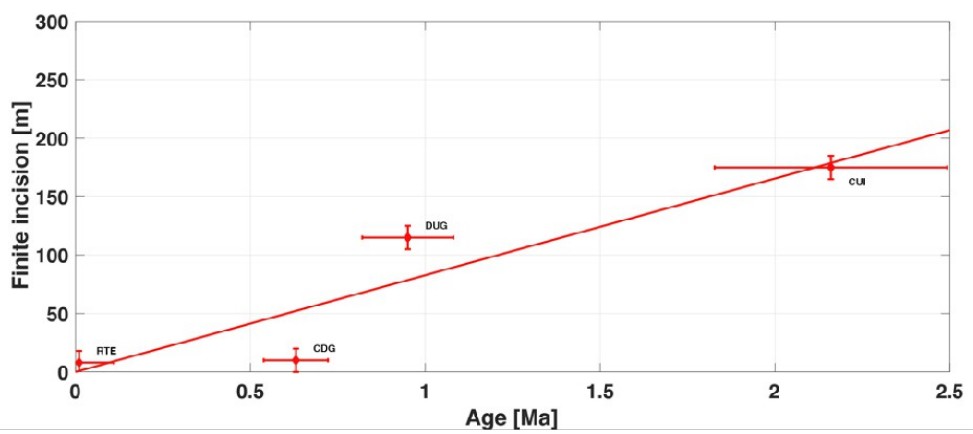

**Figure 5: Relation finite incision-burial age for the Rieutord canyon. Finite incision is the elevation of the sampling site relatively to the current riverbed. RTE for Route Cave, CDG for Camp de Guerre Cave, DUG for Dugou Cave and CUI for Cuillère Cave**



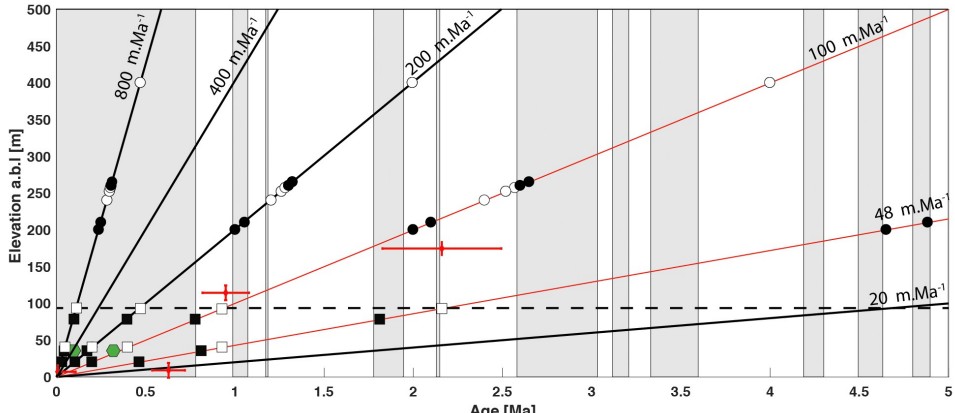

**Figure 6: Constraint on the incision rate from plural data set. Circles (Leicasse Cave system) and squares (Garrel Cave system) are paleomagnetic polarities from clay deposits. Black is for Normal polarity, white for Reversed polarity and grey for transitional signal.**
**Each point is representative of one sampling profile including an average of 10 samples per site. Lines represent different linear incision rates with example of good correlation (red ones) and bad correlation (black ones). The horizontal dashed line shows the predicted polarities-age for one site located ~ 100m a.b.l. This measured reversed polarity match with theoretical red lines but fails with the three other exposed incision-rate.**
**Green hexagons are representation of U/Th ages obtain in the Garrel (Camus, 2003). Burial ages from fig. 4 are shown for comparison (Red points)**

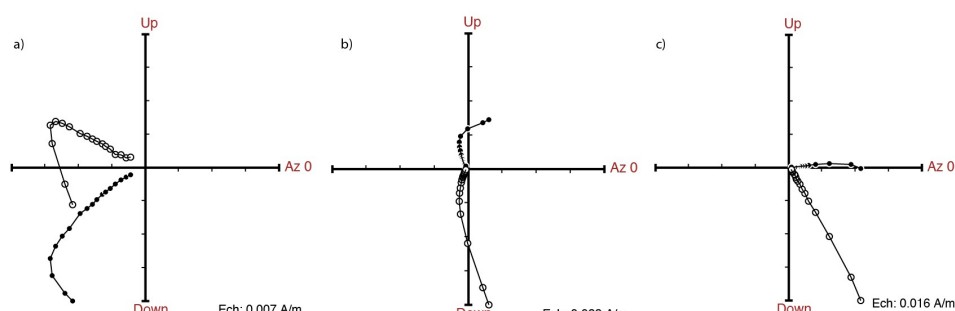

**Figure 7: Zijderveld Diagram for three samples from the Gours-sur-Pattes (Leicasse) site. Stratigraphical order is from a) (the older, base of the profile) to c) (the younger, top of the profile.**





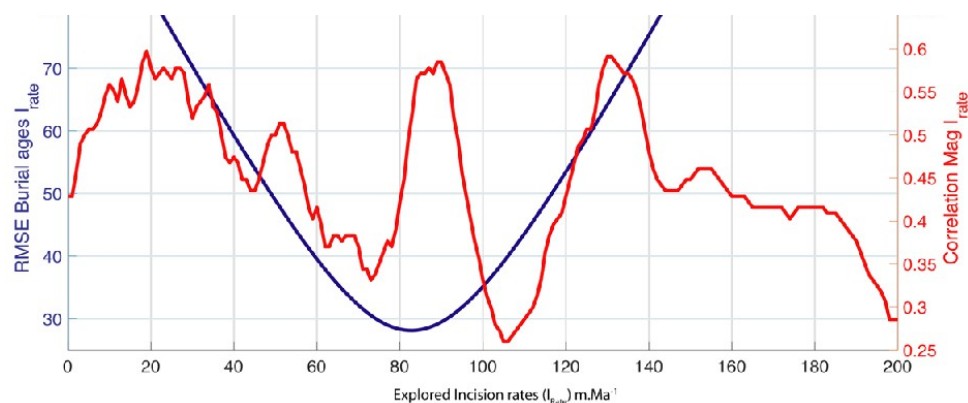

**Figure 8: Best incision rate based on paleomagnetic data (Red) and burial ages (blue). The Red curve is**
**the normalised correlation between theoretical and observed polarities. The highest correlation**
**corresponds to the best incision rates. The blue curve is the RMSE for the linear regression through the**
**burial ages data set shown on Fig. 4.**





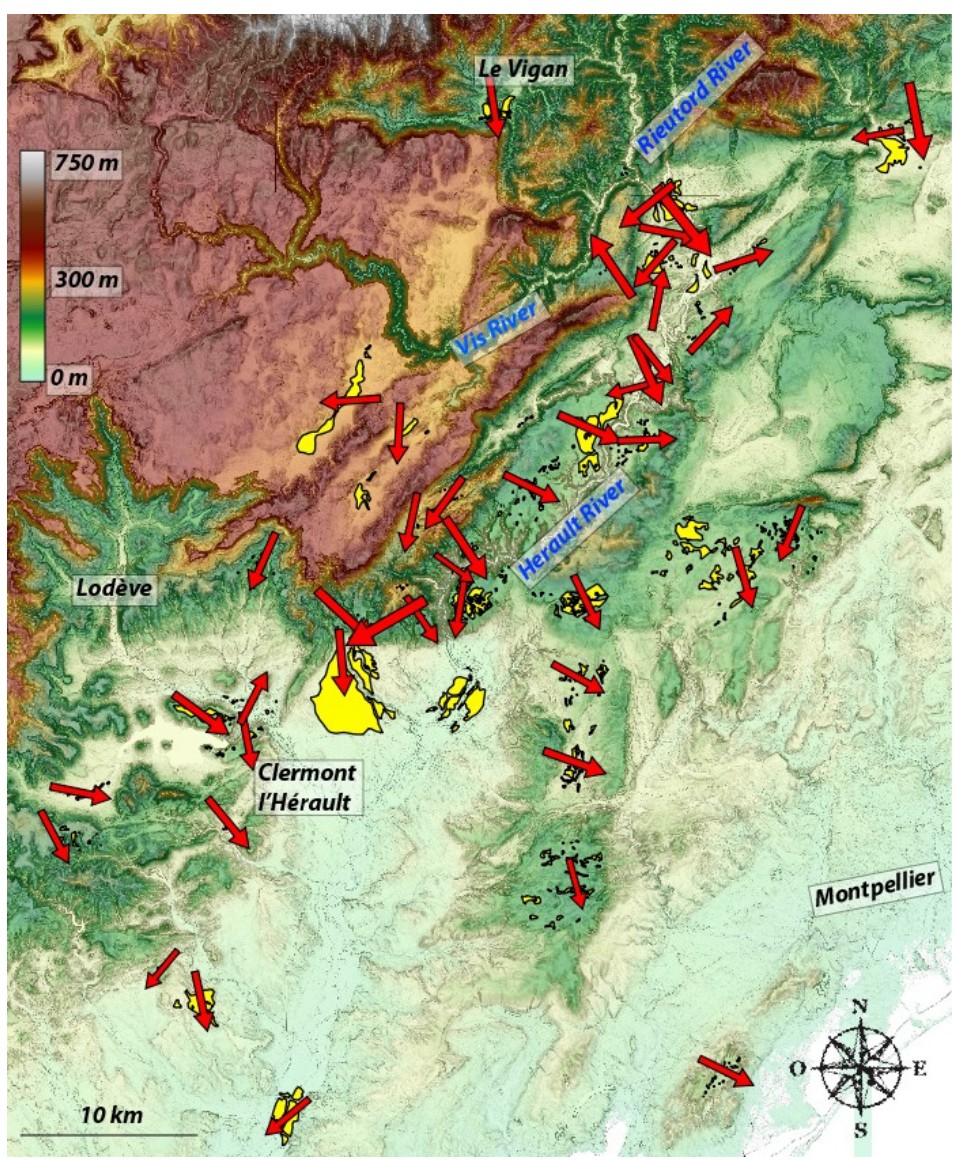

**Figure 9: Tilting map of geomorphological benchmark (yellow areas). Fond-map is 5 m resolution DEM**
**with slope shadow. Red arrows are orientating according to the marker downward dip and sized**
**according to the corrected tilting angle (the bigger, the more the tilting)**


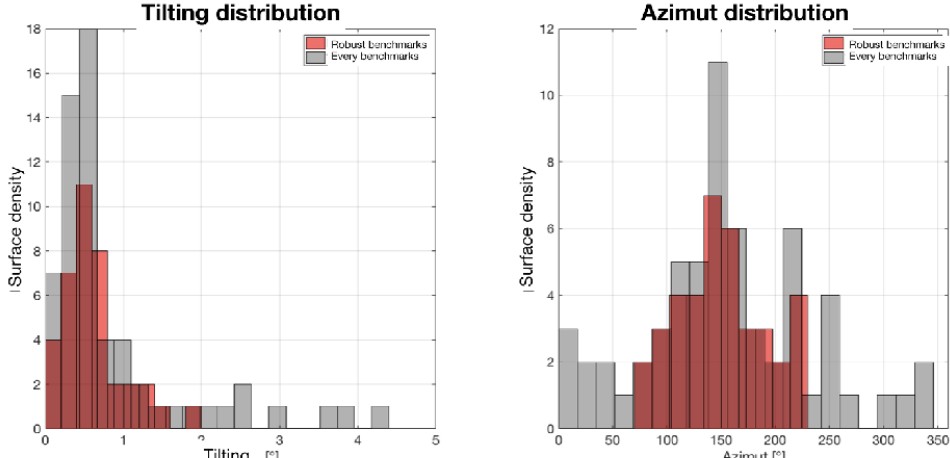

**Figure 10: Tilting and azimuth distribution. Left panel is density distribution for surface maximum**
**tilting in degree. Right panel is azimuth of maximum dipping relative to the north. For each histogram,**
**red and grey populations are for robust and primary detected markers.**

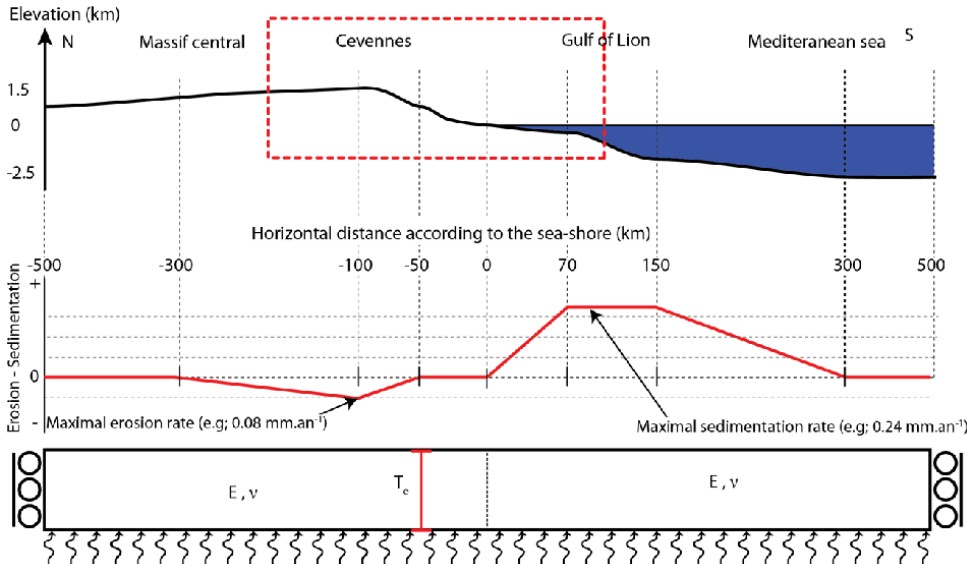

**Figure 11: Top panel: schematic topographic profile. The studied area that includes the studied zones is**
**delimited by the red box (cf fig. 1). Middle panel, surface processes profile, negative values are for erosion**
**and positive values for sedimentation. Bottom panel: model set-up with two compartments (one for the**
**Cevennes area and the second on for the gulf of lion). The base of the model is compensated in pressure**
**and the right and left limits are fixed at zero horizontal velocities and free vertical ones. Te is the**
**equivalent elastic thickness (in km), E (Pa) and ?   are the Young modulus and the Poisson coefficient**
**respectively whom values are independent in each compartment.**



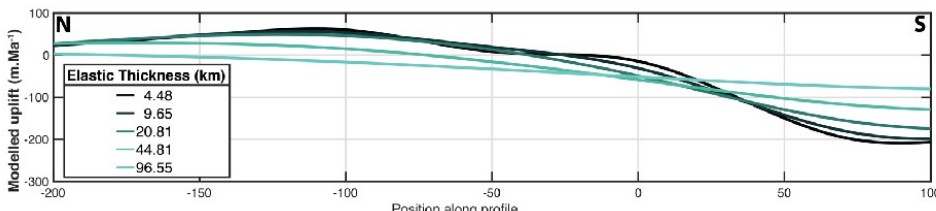

**Figure 12: Modelled uplift according to different Te. Most probable Te are between 10 and 30 km.**

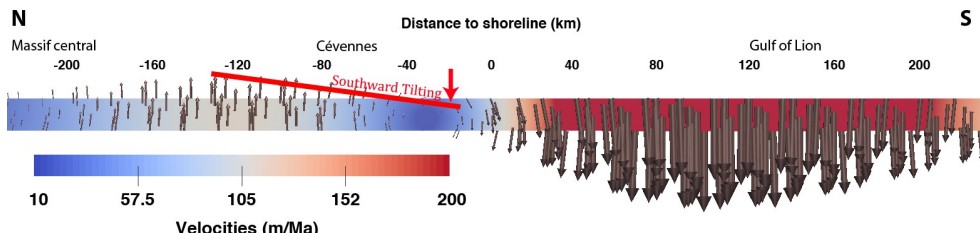

**Figure 13: Modelling result for Te= 15km. Erosion-sedimentation rate profile is the same as in fig. 6. Velocity field is shown using arrow for scale and orientation and colour code for value. Black values on top are distance relative to the sea-shore (positive value landward and negative values seaward). Red line represent the southward modelled tilting due to differential uplift.**

| Cave | Lat | Lon | Elevation | height (a.b.l) | $^{10}Be$ conc (atom/g) | $\sigma\ ^{10}Be$ (atom/g) | $^{26}Al$ conc (atom/g) | $\sigma\ ^{26}Al$ (atom/g) | $^{26}Al/^{10}Be$ (and error) | Burial age (Ma) | Burial age error (Ma) |
|---|---|---|---|---|---|---|---|---|---|---|---|
| RTE | 43,960 | 3,707 | 175 | 8 | 3,54E+04 | 1,18E+03 | 2,16E+05 | 1,47E+04 | 6,11 +/-0.46 | 0,20 | +0.16/-0.15 |
| CDG | 43,955 | 3,710 | 185 | 10 | 8,87E+04 | 3,12E+03 | 4,29E+05 | 3,28E+04 | 4,83 +/-0.41 | 0,67 | +0.18/-0.16 |
| DUG | 43,957 | 3,711 | 245 | 115 | 1,27E+04 | 5,68E+02 | 5,29E+04 | 6,36E+03 | 4,15 +/-0.53 | 0,99 | +0.28/-0.25 |
| CUI | 43,959 | 3,711 | 354 | 175 | 1,70E+04 | 7,14E+02 | 3,75E+04 | 5,28E+03 | 2,20 +/-0.32 | 2,28 | +0.33/-0.28 |

Table 1: Samples analytical results and parameters. Cave code are: RTE for the "de la route" Cave, CDG for the

"Camp de Guerre" cave, DUG for the "Dougou" Cave and CUI for the "Cuillère" Cave. Main parameters are the

geographical coordinate (Lat, Lon in decimals degree), the elevation (a.s.l), the height (a.b.l, computed

relatively to the surface river elevation. The concentration (atoms/g quartz) of 10Be and 26Al in collected sand

samples are all AMS 10Be/Be and 26Al/Al isotopic ratios  corrected for full procedural chemistry blanks and

normalised to KN-5-4  and KN -4-2, respectively.The error () is for total analytical error in final  average 10Be

and 26Al concentrations based on statistical counting error s in final  10Be/Be (26Al/Al) ratios  measured by

AMS  in quadrature with a 1% error in 9Be spike concentration  (or a 4% error in 27Al assay in quartz)   and a





2% (or 3%) reproducibility error based on repeat of AMS standards. Burial age (minimum) assuming no post-
burial production by muons at given depth (all deeper than 30m) in cave below surface and assuming initial
26Al/10Be ratio is given by the production ratio of 6.75. The burial age error determined by using a +/-1σ
range in the measured 26Al/10Be ratio