# Peer review of "Determining the Plio-Quaternary uplift of the southern French massif-Central; a new insights for"

_Solid Earth, 2019_

## Referee Comment (RC1) · Anonymous Referee #1 · 8 Jul 2019

This article from Malcles et al., presents a nice example of how cosmogenic dating of burial sediments can be used for landscape reconstruction. The multimethodological approach is particularly interesting, coupling cosmogenic and magnetostratigraphic data with geomorphological analysis and numerical model of lithospheric scale uplift.

The article is globally well written and consistently illustrated, even if as I point out below, some of the figures can be improved to provide a better understanding of the different datasets

The paper is reasonable organized the data are produced and interpreted state-of-the art and the line of arguments is generally consistent and convincing, and supports

discussion and conclusions. However, some improvement is required with respect to a couple of problems, such as the erosion trend used as input data in the numerical model.

The paper of Malcles et al. contributes substantially to reconstruct the landscape evolution and uplift history of the French Massif Central. The paper fits excellently in the profile of Solid Earth, and I would suggest publication after moderate modifications. Some general suggestions are listed below, and are complemented by specific comments in the attached pdf text file.

My main remark is related in the interpretation of the onset of the regional uplift. I find that the data well constrain the Plio-quaternary incision rates but the onset of the uplift is not well demonstrated.

Introduction The introduction does not follow a classical organization, and introduction is merged with the tectonic setting and with the list of hypothesis to prove. I do not dislike it, but I suggest to separate in a sub paragraph the discussion of the hypothesis that the authors want to test. About the three scenarios proposed I would like address your attention on the case of old uplift. The uplift could have started early and you could record only the last <5 Ma history of incision, a probably increase in the incision rates. The attached sketch explain the two alternative scenario and a possible relationship with the flat surface.

The age and the geological meaning of the flat upper surface is relevant to reconstruct the onset of uplift. Moreover, the relationship of the cave galleries with the upper flat surface and with the geomorphological markers should be better described.

Karst model One of the main point in using the cave galleries as ancient base level is to show that cave passages were really connected to the base level and that they are not only a perched level of preferable karst dissolution. For this reason I would like to see the profiles of the cave systems and its relationship with the river and eventually some photos testifying the phreatic style of the passages. Why haven't you dated

samples from the Leicasse cave, that is placed at high elevation? Higher levels exist (between 600 and 700 m) in the same region suggesting an old history of the uplift and incision. To show the sampling sites within the caves and the location of the caves on the topographic maps could help the reader.

Geomorphological analysis I found this part interesting and useful to put quantitative data in a regional scenario. I have some doubts about the paragraph organization: the three working hypothesis shown at the beginning seem a bit extraneous in a paragraph where the authors should explain how to extract the data and show the results. I suggest to rethink this organization. Also the title of the paragraph seems a bit out of context, better "analysis" instead of "evidences", for example. I wonder to see some pictures of the analyzed and discussed markers. The limit of 2° of slope is questionable. For me the problem is the topographic gradient of the entire margin. Along a NNW-SSE directed profile from Aigoual summit to the Cevennes fault the mean gradient is about 2°, with important local variations that show topographic gradient up to 4° (few kilometers SE to the summit of the Mt Aigoual for example). I would like to see if tilted plans between 2° and 4° exist. The limit of 2° is reasonable for Plio-Quaternary marker, but it is possible that older geomorphological marker could be more tilted. How the slope of each marker have been calculated?

Numerical modeling The approach is interesting and perfectly reasonable even if I am not the right person to evaluate the details. However, I have same remarks on the input data for the modeling. The authors used a regional distribution of erosion that do not correspond exactly to the published data. In figure 11 the maximum of erosion is placed at the top the Mt Aigoual, with value of 0.08 mm/yr (or 80 m/Myr) (please, change the dimension to homogenize text and figure). But, the values on top reach the minimum values testified also by the oldest thermochronological ages (long-term erosion) of Barbarand et al., 2001 and Gautheron et al., 2009. Also the cosmogenic denudation rates of Olivetti et al. 2016 suggest that the erosion on the top of the massif close to the margin is very limited (values of about 0.04 mm/yr). The values

increase along the flank, toward the lower elevation samples, confirmed by your new data set of incision rates. Therefore, the input data of the erosion distribution that the authors used for the modeling is a bit different from the measured data. I think that an erosion trend resulting bigger at lower elevation and minor at high elevation is not consistent with a process of isostatic uplift induced by erosion, supporting event more clearly the authors conclusions. I suggest to be more rigorous in the description of the geodynamic model: for instance the flexural response to the gulf of Lion extension is complicated to invoke, for the distance between the high topography and basin. The role of the mantle upwelling has been proposed by many authors (that have to be cited) that worth to be discussed a bit more in detail (dynamic or isostatically supported, the Massif Central thin lithosphere suggests, in my opinion, a clear contribution of the mantle in the present topography).

Discussion

Lines: 376 is not clear for me. The only way that I know to re-equilibrate a river profile is a regressive erosion that move from the base level upstream. If the river is full equilibrated means that regressive erosion reached the uppermost part of the profile. Moreover the lack of knickpoint does not prove that incision rate and uplift are in equilibrium, if the landscape undergoes a long topographic degradation. It could be interesting to know why the rivers profiles from northeastern margin of the Massif (Olivetti et al.) and from Ardeche (personal data) show knickpoints and Cevennes rivers not. If the authors want to discuss about the river profiles it could be interesting to show some data. Onset of volcanism is placed about 13 Ma and even earlier if we consider the synrift volcanism (Michon and Merle 2001).

Figures: In general the figures are good, but sometimes they lack of useful information such as topographic names (summits, cities, ect). I would appreciate to see the location of the analyzed caves in a map (in the figure 9 for example) and also the profile in a vertical view of the caves to have a look of the general topographic trend, its relationship with the incised river, and to show the sampling sites. Coordinates are

lacking. Figure 3 and 4 could be merged.

Please also note the supplement to this comment:
https://www.solid-earth-discuss.net/se-2019-99/se-2019-99-RC1-supplement.pdf

────────────────────────────

[Figure]

[Figure]

**Fig. 1.**

---

## Referee Comment (RC2) · Anonymous Referee #2 · 31 Aug 2019

This paper presents original data on an interesting geomorphological subject where quantification is difficult and rare. The overal conclusion that South Massif Central has seen an incision and related uplift of about 80 m/Myr in the last 4 Ma, associated with a tilt toward the south is sound and deserves publication. However, the way the data is presented is far from satisfactory (missing information, hard to understand figures, neglected data without justification, etc., see details below) and thus I suggest important revisions to be performed before acceptance. English needs also significant improvement. I point a few points below. details" 83.4+17.3/-5.4 " is too precise ! 83+17 -5 is enough. … Burial dating using Terrestrial cosmogenic nuclides (TCN) are nowadays : change are to is. Line 38 (and elsewhere): Âń can't Âń cannot is more advisable.

[Figure]

Fig.1 lacks latitude longitude and some landmarks (even myself who works in the area was not sure to locate the main structures) like main rivers, cities,... A geological map could be useful. Also the localization of studied sites is poorly precised in this figure. Could they be also indicated in e.g. fig.9? Line 123-124: sentence needs a verb! Line 166-169 : strange practice to give results in the methods section (2.1) Please move them to section 2.2! Line 83±35 is enough precise. A table of paleomagnetic results with statistical parameters is mandatory. Fig.6 is hard to understand (especially not knowing how much paleomagnetic sites are available). I figure that on paleomag polarity is represented arbitrarily by a set of points fitted with chosen incision rate, allowing to see if the polarity is consistent with the scale, indicated as vertical grey strips. This is very badly explained! Line 219 "First, we note a good agreement between samples located at the same elevation," I really don't get how you derive such assertion! Line 223-225: about this reverse-normal sequence, there is no way to see it on Fig.6! Again the table is mandatory! You have to comment on the reverse polarity at ≈40 m that you assign to Brunhes period. Why not putting Matuyama there? Line 243: "Using a similar approach for the Rieutord crystalline samples," I don't get what you mean! How do you compute average dip and azimuth of your geomorphological surfaces? If it's arithmetic mean, that not acceptable. You have to make it using directional statistics (and show us a stereogram of dip lines) Is Fig.9 all markers or only the robust ones? The second option (38 data; but I count 45 on fig.9!) seems right. But then the azimuths exhibit in fig.9 does not fit Fig.10. There are northward dips! Fig.10 scale "surface density" is a number of surfaces? Please make this clear.

---

## Author Comment (AC1) · 27 Sep 2019

Modified manuscript is provided in supplement

Reviewer 1

This article from Malcles et al., presents a nice example of how cosmogenic dating of burial sediments can be used for landscape reconstruction. The multi-methodological approach is particularly interesting, coupling cosmogenic and magnetostratigraphic data with geomorphological analysis and numerical model of lithospheric scale uplift. The article is globally well written and consistently illustrated, even if as I point

out below, some of the figures can be improved to provide a better understanding of the different datasets The paper is reasonable organized the data are produced and interpreted state-of-the art and the line of arguments is generally consistent and convincing, and supports discussion and conclusions. However, some improvement is required with respect to a couple of problems, such as the erosion trend used as input data in the numerical model. The paper of Malcles et al. contributes substantially to reconstruct the landscape evolution and uplift history of the French Massif Central. The paper fits excellently in the profile of Solid Earth, and I would suggest publication after moderate modifications. Some general suggestions are listed below, and are complemented by specific comments in the attached pdf text file.

Q1: My main remark is related in the interpretation of the onset of the regional uplift. I find that the data well constrain the Plio-Quaternary incision rates but the onset of the uplift is not well demonstrated.

A1: Although we agree with the reviewer #1, the time period covered by our samples is unfortunately following the onset of the uplift and any conclusion on the onset of the uplift would be nothing more than an hypothesis loosely supported by the data. For instance, the area is part of the lithospheric-scale structure which is the Massif Central and it makes sense that the local evolution is linked to the regional evolution. We agree that this question is important to address for a better understanding of the regional dynamic, and hence, for stable continental area deformation driving processes, but it is out of the scope of the paper. We made it clearer in the paper. Line 59-60: adding "Further studies should aim to address the problem of uplift onset, giving more clues concerning the stable continental area but owing the data we presently have, discussing such onset is out of the scope of the paper."

Q2: Introduction The introduction does not follow a classical organization, and introduction is merged with the tectonic setting and with the list of hypothesis to prove. I do not dislike it, but I suggest to separate in a sub paragraph the discussion of the hypothesis that the authors want to test.

[Figure]

A2: In order to make it clear, we split the Introduction in two parts: 1.1 Introduction (line 36) and 1.2: Working hypothesis (line 95)

Q3: About the three scenarios proposed I would like address your attention on the case of old uplift. The uplift could have started early and you could record only the last <5 Ma history of incision, a probably increase in the incision rates. The attached sketch explain the two alternative scenario and a possible relationship with the flat surface.

A3: First, we agree that the incision-rate is probably not linear if looking at higher frequencies, and could have started earlier. For example, such incision-rate variations have been proposed for the Alps (See Saillard et al., 2014; Rolland et al., 2017; Line 386). There is not any possibility to test, based on our data, if there was an increase in the last 5 Myrs, but we can not rule out a marginal increase over the last 5Myrs. Concerning the flat surfaces, we understand this comment as rising the question of possible apparent dip due to diffusion processes and not due to differential uplift. First, we point out that surfaces present southward dipping on both river-sides (if the dipping is due to diffusion processes, we expect dipping toward the river). Second, diffusion processes will mark the surfaces edges first, creating convex topography. This convex shape (and probable increasing dip) would be discarded either by automatic recognition or by manual control of the surface robustness. To take into account this remark we added for clarity: " Diffusion processes could create apparent tilt of remnant horizontal surfaces. However, we avoid that problem by completing the automatic selection and correction with a final check to make sure that the residuals are randomly distributed over the surface (see below)." (Lines 283-285).

Q4: The age and the geological meaning of the flat upper surface is relevant to reconstruct the onset of uplift.

A4: Agree, but because of the strong uncertainties concerning the ages of the upper surface (e.g. possible important time-lag between the surface formation and the uplift onset), we chose not to discuss it in the paper that deals mainly with the Plio-

Quaternary dynamic.

Q5: Moreover, the relationship of the cave galleries with the upper flat surface and with the geomorphological markers should be better described. Karst model One of the main point in using the cave galleries as ancient base level is to show that cave passages were really connected to the base level and that they are not only a perched level of preferable karst dissolution. For this reason I would like to see the profiles of the cave systems and its relationship with the river and eventually some photos testifying the phreatic style of the passages.

A5: We agree with possible alteration-driven karstification but, we do not propose to discuss the cave formation. Indeed, to our knowledge, there is no way to date the void created by karst galleries, therefore we use them only as empty pockets trapping sediments without any doubt, and now located on canyon wall as already done by Granger et al., (1997, 2001). This is the case at least for the Rieutord caves and the Garrel, for the Leicasse cave we refer the reader to Camus (2003) where the link to fluvial transport has been shown. Therefore, we think that adding more pictures of the caves and figures would not bring a significant gain in clarity. We refer the reader to publication where these relations are discussed e.g. Audra et al., (2001) or Moccochain (2007) and the link to fluvial transport has been shown (Camus, 2003). To allow the reader to have a look at the geometry of the caves we added a link to a university hosted database with a doi where the caves are available in 3D. Line 1548-154: Karst3D (2019). Karst3D data base. https://doi.org/10.15148/940c2882-49f1-49db-a97e-12303cace752

Q6: Why haven't you dated samples from the Leicasse cave, that is placed at high elevation? Higher levels exist (betwen 600 and 700 m) in the same region suggesting an old history of the uplift and incision.

A6: Dating of other samples is in progress and should be subject to publication when obtained. Furthermore, we point out that to our knowledge quartz bearing infilling are not present at 600-700m a.s.l. In the explored caves. Highest known quartz cobbles in the Leicasse cave are located c.a. 450 a.s.l.

Q7: To show the sampling sites within the caves and the location of the caves on the topographic maps could help the reader.

A7: See above for topographic survey database. (Sampling site will be added).

Q8: Geomorphological analysis I found this part interesting and useful to put quantitative data in a regional scenario. I have some doubts about the paragraph organization: the three working hypothesis shown at the beginning seem a bit extraneous in a paragraph where the authors should explain how to extract the data and show the results. I suggest to rethink this organization.

A8: The paragraph structure has been rewritten in order to make it clear with in order: methods, expectations and results. Line 257 to 267 have been moved to Line308-320.

Q9: Also the title of the paragraph seems a bit out of context, better "analysis" instead of "evidences", for example.

A9: Changed to "Geomorphometrical approach" Line 276.

Q10: I wonder to see some pictures of the analyzed and discussed markers.

A10: We added as supplementary some pictures of the discussed flat surfaces.

Q11: The limit of 2âŮę of slope is questionable. For me the problem is the topographic gradient of the entire margin. Along a NNW-SSE directed profile from Aigoual summit to the Cevennes fault the mean gradient is about 2âŮę, with important local variations that show topographic gradient up to 4âŮę (few kilometers SE to the summit of the Mt Aigoual for example). I would like to see if tilted plans between 2âŮę and 4âŮę exist. The limit of 2âŮę is reasonable for Plio-Quaternary marker, but it is possible that older geomorphological marker could be more tilted.

A11: We agree that using $2°$ as slope cut-off will limit the detection of some surfaces and lead to miss some markers. But we focus on the Plio-Quaternary evolution, therefore the 2° cut-off could even be seen as a proxy for older (and polyphased) surface filter. For instance, old surface in the Larzac plateau have been proposed by Bruxelles (2001) but we didn't include them in our analysis because of their expected old ages and so forth, possible strong alterations, plural-deformation registration, etc.

Q12: How the slope of each marker have been calculated?

A12: As explained in lines 280-302 we use automatic and manual delimitation of surface, iterative plan fitting using extracted DEM points and statistical outliers suppression, and robustness criterions filter. We have modified these lines to make it clearer.

Q13: Numerical modeling The approach is interesting and perfectly reasonable even if I am not the right person to evaluate the details. However, I have same remarks on the input data for the modeling. The authors used a regional distribution of erosion that do not correspond exactly to the published data. In figure 11 the maximum of erosion is placed at the top the Mt Aigoual, with value of 0.08 mm/yr (or 80 m/Myr) (please, change the dimension to homogenize text and figure). But, the values on top reach the minimum values testified also by the oldest thermochronological ages (long-term erosion) of Barbarand et al., 2001 and Gautheron et al., 2009. Also the cosmogenic denudation rates of Olivetti et al. 2016 suggest that the erosion on the top of the massif close to the margin is very limited (values of about 0.04 mm/yr). The values increase along the flank, toward the lower elevation samples, confirmed by your new data set of incision rates.

A13: Dimensions in figure have been changed for consistency. We extended the explanations in the text to explain the erosion-profile setting (lines 352-360): "This profile is a simplification of the one that can be expected from Olivetti et al. (2016) and do not aim at matching precisely the published data because of, first, the explored time-span ($\sim$ 1 Myrs) is not covered by thermochronological data (> 10Myrs) or cosmogenic denudation rate (10s-100s kyrs). Second, we assume that erosion rates are correlated to the first order to the local (10s km2) slopes, that are higher near the drainage divide. This allows to include any kind of erosion processes (e.g. landslides). Third, the model supposes a cylindrical structure perpendicular to the cross section, this implies to average the high-frequency lateral variations of slope, elevation, etc. to derive the actual denudation rate based on these proxies. Concerning this erosion profile, a parametric study with highest erosion rate ranging from 1 to 1000 m.Myrs-1 led to the same first order interpretations.."

Q14: Therefore, the input data of the erosion distribution that the authors used for the modeling is a bit different from the measured data. I think that an erosion trend resulting bigger at lower elevation and minor at high elevation is not consistent with a process of isostatic uplift induced by erosion, supporting event more clearly the authors conclusions.

A14: Using numerous erosion profiles, but with always a pattern of erosion on the relief and deposit off-shore won't change the first order results since they are mainly controlled by the Elastic parameters. We extended description of the choice of the erosion profile to make it clear (lines 352-360. (See A13 above).

Q15: I suggest to be more rigorous in the description of the geodynamic model: for instance the flexural response to the gulf of Lion extension is complicated to invoke, for the distance between the high topography and basin. The role of the mantle upwelling has been proposed by many authors (that have to be cited) that worth to be discussed a bit more in detail (dynamic or isostatically supported, the Massif Central thin lithosphere suggests, in my opinion, a clear contribution of the mantle in the present topography).

A15: We agree that mantle or deep processes are involved in the uplift onset. However, we developed a conceptual to test the role of the erosion-induced isostatic adjustment and not to elaborate more complicated models with too many parameters compared to the constraints that we have. For instance, we cannot decipher if the thin and slightly hot lithosphere is related to dynamic topography with ongoing mantle upwelling or if it is supported by thermal isostasy being a remnant of past processes like the opening of the Gulf of Lion.

Discussion Q16: Lines: 376 is not clear for me.

A16: "Hence, the incision rate has to be balanced to the first order by the uplift rate". Explanation has been added for better clarity Lines 420 – 424: "Hence, back to the three conceptual models presented in part 1 (Fig.2), we can discard, at first order, the models A (Old uplift-recent incision) and B (Old uplift-old incision) because the obtained incision rate shows recent incision and surface tilting tend to prove a current uplift. Therefore, the incision rate has to be balanced to the first order by the uplift rate. Eustatic variations magnitudes are of too low (100-120 m) to explain the total incision (up to 400m). "

Q17: The only way that I know to re-equilibrate a river profile is a regressive erosion that move from the base level upstream. If the river is full equilibrated means that regressive erosion reached the uppermost part of the profile. Moreover the lack of knickpoint does not prove that incision rate and uplift are in equilibrium, if the landscape undergoes a long topographic degradation.

A17: We agree that the lack of knickpoints do not prove an equilibrium state (and that in general the term "equilibrium" is subject to debate) but it allows to dismiss a strong impact of regressive erosion due to recent sea-level variations or tectonic. To address the concern of long topographic degradation (assuming no uplift?), we point out that such degradation will lead to mass export, then lithospheric unloading and then isostatic-adjustment uplift, which is why we developed our conceptual model to test if this process could be responsible for our observations.

Q18: It could be interesting to know why the rivers profiles from northeastern margin of the Massif (Olivetti et al.) and from Ardeche (personal data) show knickpoints and Cevennes rivers not. If the authors want to discuss about the river profiles it could be interesting to show some data.

A18: This subject looks important and further study should be addressed in this sense. But discussion about the river profiles are beyond the scope of this paper and seems to us unnecessary, especially in a manuscript that is already long and complex. Indeed, for drawing robust conclusions, we should, as suggested by Reviewer #1 enlarge the study to other rivers surrounding the Massif Central and not only the ones in our study. As Reviewer #1we noticed the difference in knickpoints between the southern and eastern margin of the Massif-Central and we think that ot deserve a more complete study, notably to discuss a possible role of karstic dynamic given the major lithological difference between these two regions.

Q19: Onset of volcanism is placed about 13 Ma and even earlier if we consider the synrift volcanism (Michon and Merle 2001).

A19: We agree, according to Nehlig et al., the volcanism is started 65 Ma ago in some places. One of the many issues they highlighted was the diachronism of the volcanic activity throughout the Massif-Central. For example, they shown an activity spanning from 13 to 2Ma with a paroxysm at 8.5 Ma for the Cantal stratovolcano. Dautria et al. (2010) proposed younger ages of volcanic structures southward, etc. We chose 5 Ma as an average of increased activity throughout the area but not as the onset of the volcanisms which is on the other hand is of major importance in the discussion of the uplift onset, as previously discussed.

Q20: Figures: In general the figures are good, but sometimes they lack of useful information such as topographic names (summits, cities, ect). I would appreciate to see the location of the analyzed caves in a map (in the figure 9 for example) and also the profile in a vertical view of the caves to have a look of the general topographic trend, its relationship with the incised river, and to show the sampling sites.

A20: Changed accordingly with some geographical information, trying not to overload the figures. See modified figure.

Q21: Coordinates are lacking.
A21: Lacking coordinates are now provided in the figure 1 caption.

Figure 3 and 4 could be merged. Please also note the supplement to this comment: https://www.solid-earth-discuss.net/se-2019-99/se-2019-99-RC1-supplement.pdf Modifications included, and answer in the section below.

Supplementary review 1:

SQ1: Line 43: I agree with the recent uplift, but it is still under debate. The topography could be also interpreted as a longlasting degradation of an ancient topography.

SA1: We agree. This hypothesis is presented latter in this section. We changed the sentence in order to present it as our chosen hypothesis but not the unique one possible.

SQ2: Line 58: 9 kms seem a bit too much, anyway big thickness is found in the depocenter basin, such as the center of Ales Basin, while along the Cevennes margin the thickness progressively decrease toward the NW. It is not proven that the entire Cevennes region was covered by Mesozoic sediments.

SA2: Noted and slight changes in the sentence ("reach several kilometers" instead of "be more than 9km". The 9 kms are for the overall SE basins (southward of Ales Basin). Anyway, exact spatial coverage or thickness will not change our point given that the first order is sufficient to our study.

SQ3: Line 60: The uplift event is called Durancian uplift event while the Isthmus is the topographic high formed as a consequence, I think.

SA3: Noted and changed accordingly.

SQ4: Line 69: do you refer to Sanchis and Seranne?

SA4: Indeed, as an example of evolution induced by the extensional period, not as direct study of the watershed evolution.

[Figure]

SQ5: Line 70: To be meticulous, the events are three: the Mid-cretaceous uplift, the Pyrenean compression and the Oligocene extension.

SA5: We agree. The Durancian event (Mid-Cretaceous uplift) is presented before in the section but should mentioned here. Changed accordingly.

SQ6: Line 93:

SA6: Tilt –> Tilting

SQ7: line 55-56: I suggest to use Ma instead of Myrs ago

SA7: Changed accordingly.

SQ8: Line 132: It could be useful for the readers a briefly description of the morphological setting of the area, with the plateau, canyons ect.

SA8: Short descriptions are provided Line 54 to 59. We added call to figure 1 (for the provided topographic map) for visual insights into first order morphology.

SQ9: Line 136: sequence

SA9: Changed accordingly

SQ10: Line 140:

SA10: References writing changed

SQ11: Line 154:

SA11: Some precisions concerning the sediments protolith area are now provided.

SQ12: Line 594: fig. 9.

SA12: Changed accordingly

SQ13: Figure 1: It is not very clear the location of the Massif Central. It could be interesting to see the sampling location divided for methods, where you performed cosmogenic analysis and where paleomagnetism.

SA13: Context map was changed with zoom over western Europe. More information added into the figure caption. See revised figure.

SQ14: Figure 11: change mm.yr-1 in m.Myr-1 "the studied area that include the studies zones" sounds a bit as tautology. It could be better to clarify. the simbol v is gone

SA14: Dimensions changed for consistency to m.Ma-1. Unclear sentence was removed. Symbol v has been added.

Please also note the supplement to this comment:
https://www.solid-earth-discuss.net/se-2019-99/se-2019-99-AC1-supplement.pdf

**Supplement:**

[revised manuscript text omitted]

on the karst network geometry and its relation to morphological markers can be find in Camus (2003).
Dating accurately the emplacement of cave galleries is a challenge and beyond the scope of this paper.
We only aim at using the dating information brought by sediment that have been trapped into the cave
system. Therefore, we apply the classical model (e.g. Harmand et al., 2017) where sediments in cave
opened on canyon walls are related to terraces or other geomorphologic markers. In this kind of
context, former studies suggest that the hypotheisis is valuable (i.e. Granger et al., 1997, 2001).  For
cave topographic surveys, we refer the reader to the Karst3D database (Karst3D, 2019).

[revised manuscript text omitted]
 where we minimize the residual between the observed and the modeled ages based on the same incision-rate range than for the paleomagnetic samples. With this method, we determined a linear incision rate of $85 \pm 11$ m.Ma$^{-1}$ (Fig 8). Those two results, based on independent computations, suggest the same first-order incision rate for the last 4 Ma of 84 $^{+21}/_{-12}$ m.Ma$^{-1}$. Given that the Rieutord, Garrel and Buèges rivers are all tributaries of the Hérault river, we propose that this rate represents the incision rate for the Hérault river watershed, inducing approximately 300-350 m of finite incision over the Plio-Quaternary period. If the landscape is at first order in an equilibrium state, that is to say, if we preclude our incision rates being a regressive erosional signal, the incision needs to be balanced by an equivalent amount of uplift. If the uplift rate is roughly correlated to the regional topography, lowest uplift rates would be expected in the south of our sampling sites inducing regional tilting of morphological benchmarks. In the next part, we search for such evidences that would suggest differential uplift.

**2.3 **Geomorphometrical approach**

According to the Massif-Central centered uplift hypothesis, morphological markers such as strath terraces, fluvio-karstic surfaces or abandoned meanders should display a southward tilting due to differential uplift between the northern and the southern part of the region.

To investigate these different signals, we used the morphological markers available for the study area (Fig. 9). We used a 5 m resolution DEM analysis to identify the markers corresponding to surfaces with slope < 2°. This cut-off slope angle prevents to identify surface related to local deformation such as for example landslide or sinkhole. Diffusion processes could create apparent tilt of remnant horizontal surfaces. However, we avoid that problem by completing the automatic selection and correction with a final check to make sure that the residuals are randomly distributed over the surface (
[revised manuscript text omitted]
) The red box shows the Location of the study area and fig. 9. Numbers indicate sampling sites; TCN measurements: 1) Rieutord Canyon (43,958°N; 3.709°E); Paleomagnetic analysis: 2) Leicasse Cave System (43,819°N; 3.56°E) and 3) Garrel Cave system (43,835°N; 3.616°E); dated basaltic flows: 4) Lodève basin (43,669°N; 3.382°E). Bottom panel is an example of typical topographic profile used for numerical model set up, its location is given by the blue line on the map.**
**Note the north-western area with large limestone plateaus dissected by canyons (Grands Causses), and the rugged crystalline area with steep valleys (Cévennes).**

[Figure]

**Figure 2: conceptual models for landscape evolution. Top panel is the initial stage (prior to uplift). Each panel represent a possible scenario explaining current morphology: A) Old uplift and old incision, B) Old uplift and recent incision and C) both recent uplift and incision. Blue arrow and associated ages show expected result (or absence of) for burial dating. Red level represents fossilized morphological markers, cumulating (or not) the differential uplift by finite tilting.**

[Figure]

**Figure 3: Example of quartz cobbles sampled for burial dating. Location: Cuillère Cave**

[Figure]

**Figure 4: Example of clay sampling for the paleomagnetic study. Location at the entrance shaft of the Le-**
**icasse Cave system. With a ~580 m a.s.l. elevation, they are the highest samples.**

[Figure]

**Figure 5: Relation finite incision-burial age for the Rieutord canyon. Finite incision is the elevation**
**of the sampling site relatively to the current riverbed. RTE for Route Cave, CDG for Camp de Guerre**
**Cave, DUG for Dugou Cave and CUI for Cuillère Cave**

[Figure]

Figure 6. Constraining the incision rate in the Cevennes margin, using paleomagnetic polarities from clay deposits (black, grey and white symbols) and burial ages (red crosses): Circles are from the Leicasse cave with LGP being *les gours sur pattes profile (see text)*, squares are from the Garrel cave. Black, grey and white symbols correspond to normal, transitional and reverse polarities, respectively. Black linear straight lines define possible incision rates that are supposed constant over the study time period. Cf values are the correlation factors between the measured paleomagnetic polarities and the predicted paleomagnetic scale (see also Figure 8). Green hexagons show the U/Th ages obtained in the Garrel by Camus (2003).

[Figure]

Figure 7: Zijderveld Diagram for three samples from the Gours-sur-Pattes (Leicasse) site. Stratigraphical order is from a) (the older, base of the profile) to c) (the younger, top of the profile.

**Figure 8: Linear incision rate fit to the paleomagnetic data (blue) and TCN burial ages (red). The**
**blue curve is the normalized correlation between theoretical and observed polarities. The highest**
**correlation corresponds to the best incision rates. The red curve is the RMSE between the modeled**
**and the observed burial ages shown on Fig. 4, the lower the RMSE, the better the fit.**

[Figure]

**Figure 9: Color-coded-altitude map based on a 30 m resolution DEM with slope shading with tilted**
**geomorphologic surfaces indicated by blue polygons. . The vectors show the marker dip direction**
**and are proportional to the tilt angle (the larger vectors are for higher tilts). Yellow and brown**
**vectors are for robust and rejected surfaces respectively. Several vectors are hidden due to their**
**close proximity to the larger ones. Numbers indicate the sampling sites: 1) Rieutord Canyon, 2)**
**Leicasse Cave System, 3 Garrel Cave system and 4) Lodève basin. See Fig. 1 for geographic**
**coordinates.**

[Figure]

**Figure 10: Tilt and azimuth distribution. Left panel is density distribution for surface maximum**
**tilt in degree. Right panel is the dip azimuth relative to the north. For each histogram, red and grey**
**populations are for robust and primary detected markers.**

[Figure]

**Figure 11: Top panel: simplified topographic profile. The red box corresponds to the area shown**
**on figs. 1 and 9. Mid panel, surface processes profile, negative values are for erosion and positive**
**values for sedimentation. Bottom panel: model set-up with two compartments (one for the**
**Cevennes area and one for the gulf of lion). The base of the model is supported by hydrostatic**
**pressure and the right and left boundaries are free to move vertically but their horizontal**
**velocities are set to 0 mm/yr. Te is the equivalent elastic thickness (in km), E (Pa) and v are the**
**Young modulus and the Poisson coefficient respectively whom values are independent in each**
**compartment.**

[Figure]

**Figure 12: Modelled uplift according to different elastic thickness (Te). Most probable Te are between 10 and 30 km.**

[Figure]

**Figure 13: Modelling result for Te= 15km. Erosion-sedimentation rate profile is the same as in fig. 9. The vectors show the velocity field and the intensity is given by the background color code. Black values on top are distance relative to the sea-shore (positive value offshore and negative values inland). The red line represents the southward modelled tilting due to differential uplift.**

| Cave | Lat | Lon | Elevation | height (a.b.l) | 10Be conc (atom/g) | σ 10Be (atom/g) | 26Al conc (atom/g) | σ 26Al (atom/g) | 26Al/10Be (and error) | Burial age (Ma) | Burial age error (Ma) |
|---|---|---|---|---|---|---|---|---|---|---|---|
| RTE | 43,960 | 3,707 | 175 | 8 | 3,54E+04 | 1,18E+03 | 2,16E+05 | 1,47E+04 | 6,11 +/-0.46 | 0,20 | +0.16/-0.15 |
| CDG | 43,955 | 3,710 | 185 | 10 | 8,87E+04 | 3,12E+03 | 4,29E+05 | 3,28E+04 | 4,83 +/-0.41 | 0,67 | +0.18/-0.16 |
| DUG | 43,957 | 3,711 | 245 | 115 | 1,27E+04 | 5,68E+02 | 5,29E+04 | 6,36E+03 | 4,15 +/-0.53 | 0,99 | +0.28/-0.25 |
| CUI | 43,959 | 3,711 | 354 | 175 | 1,70E+04 | 7,14E+02 | 3,75E+04 | 5,28E+03 | 2,20 +/-0.32 | 2,28 | +0.33/-0.28 |

Table 1: Samples analytical results and parameters. Cave code are: RTE for the "de la route" Cave, CDG for the "Camp de Guerre" cave, DUG for the "Dugou" Cave and CUI for the "Cuillère" Cave. Main parameters are the geographical coordinate (Lat, Lon in decimals degree), the elevation (a.s.l), the height (a.b.l., computed relatively to the surface river elevation. The concentration (atoms/g quartz) of 10Be and 26Al in collected sand samples are all AMS 10Be/Be and 26Al/Al isotopic ratios corrected for full procedural chemistry blanks and normalized to KN-5-4 and KN -4-2, respectively. The error () is for total analytical error in final average 10Be and 26Al concentrations based on statistical counting error s in final 10Be/Be (26Al/Al) ratios measured by

AMS  in quadrature with a 1% error in 9Be spike concentration  (or a 4% error in 27Al assay in quartz)   and a

2% (or 3%) reproducibility  error based on repeat of AMS standards. Burial age (minimum) assuming no post- burial production by muons at given depth (all deeper than 30m) in cave below surface and assuming initial

26Al/10Be ratio is given by the production ratio of 6.75. The burial age error determined by using a  +/-1σ range in the measured 26Al/10Be ratio

---

## Author Comment (AC2) · 27 Sep 2019

Modified manuscript is provided in supplement.

Q1: This paper presents original data on an interesting geomorphological subject where quantification is difficult and rare. The overall conclusion that South Massif Central has seen an incision and related uplift of about 80 m/Myr in the last 4 Ma, associated with a tilt toward the south is sound and deserves publication. However, the way the data is presented is far from satisfactory (missing information, hard to understand figures, neglected data without justification, etc., see details below) and thus I suggest important revisions to be performed before acceptance. English needs also

significant improvement. I point a few points below.

A1: We took into consideration your comments and suggestions. We hope that our revisions and our answers below will clarify our work.The English has been reviewed. We would like to emphasis that one of the author is a native English speaker (Australia).

Q2: details" 83.4+17.3/-5.4 " is too precise ! 83+17 -5 is enough. . ..

A2: Agree, and change accordingly.

Q3: Burial dating using Terrestrial cosmogenic nuclides (TCN) are nowadays: change are to is. Line 38 (and elsewhere): Ân ÌĄ can't Ân ÌĄ cannot is more advisable.

A3: Corrected

Q4: Fig.1 lacks latitude longitude and some landmarks (even myself who works in the area was not sure to locate the main structures) like main rivers, cities,. . .

A4: Additional informations have been added to the map, trying to fin a balance between information and clarity. See revised figure.

Q5: A geological map could be useful.

A5: At the scale of the study region and given the topic of the manuscript, we think that a topographic map is much more useful than a geological map to observe the overall morphology (besides, we mentioned clearly in the introduction and the tectonic setting that the geology of the studied area corresponds mainly to Mesozoic to lower Cenozoic limestones plateaus).

Q6: Also the localization of studied sites is poorly precised in this figure.

A6: Given the scale of Figure 1, it is not possible to locate precisely the studied points. We gave their precise geographical coordinates in the figure captions.

Q7: Could they be also indicated in e.g. fig.9?

A7: Yes, we added them.

**SED**

Q8: Line 123-124: sentence needs a verb!

A8: "Creating" changed to "creates".

Q9: Line 166-169 : strange practice to give results in the methods section (2.1) Please move them to section 2.2!

A9: We did it (see revised manuscript lines 211-214)

Q10: Line 216: 83±35 is enough precise. A table of paleomagnetic results with statistical parameters is mandatory. A10: We added the table in the supplementary material.

Q11: Fig.6 is hard to understand (especially not knowing how much paleomagnetic sites are available). I figure that on paleomag polarity is represented arbitrarily by a set of points fitted with chosen incision rate, allowing to see if the polarity is consistent with the scale, indicated as vertical grey strips. This is very badly explained !

A11: Almost all the part 2.2.2, the figure and the captions have been reworked for better clarity.

Q12: Line 219 "First, we note a good agreement between samples located at the same elevation," I really don't get how you derive such assertion!

A12: We added explanation Line 2541-242: "samples located at the same elevation and being part of the same stratigraphic layer (Camus, 2003). This syngenetic deposition allow, as best explanation to prevent from a possible partial endokarstic reworking". Indeed, Some sampling sites are located at slightly different elevation but inside the same gallery and, as part of the same sedimentary layer they have to display the same polarity, that is the case. This consistency is not on its own a proof that the clay didn't sediment in different period (with same polarity) but it is by far the most reasonable explanation. See supplementary material with analysis details.

Q13: Line 223-225: about this reverse-normal sequence, there is no way to see it on Fig.6! Again the table is mandatory! You have to comment on the reverse polarity at

≈40 m that you assign to Brunhes period. Why not putting Matuyama there?

A13: Given the poor quality of the data, we skipped it from the interpretation. See supplementary material.

Q14: Line 243: "Using a similar approach for the Rieutord crystalline samples," I don't get what you mean!

A14: We added simple explanations in the paragraph: "Using a similar approach for the Rieutord crystalline samples where we minimize the residual between the observed and the modeled ages based on the same incision-rate range than for the paleomagnetic samples".

Q15: How do you compute average dip and azimuth of your geomorphological surfaces? If it's arithmetic mean, that not acceptable. You have to make it using directional statistics (and show us a stereogram of dip lines)

A15: Given values are indeed not arithmetic mean. We checked again the computation and minor errors have been fixed (Average dip changed from 0.61 ± 0.41° with an azimuth of N150 ± 40°E tot 0.60 ± 0.40° with an azimuth of N128 ± 36°E). It does not change the interpretation. Errors was due to mistake in downward sign for 2 surfaces. Because of very low dip angle of the plane, the conventional representation through stereogram is useless and won't provide the information brought by the histograms.

Q16: Is Fig.9 all markers or only the robust ones? The second option (38 data; but I count 45 on fig.9!) seems right. But then the azimuths exhibit in fig.9 does not fit Fig.10. There are northward dips!

A16: Fig. 9 doesn't display only the robust values. We added different color in order to decipher in between the two sets. Note that some surfaces cannot be shown on the map because of their small size and their closeness. See revised figure 9.

Fig.10 scale "surface density" is a number of surfaces? Please make this clear.

A10: Yes. Modified accordingly.

Please also note the supplement to this comment:
https://www.solid-earth-discuss.net/se-2019-99/se-2019-99-AC2-supplement.pdf

---

## Author Response (AR2)

Dear editor.

Thank you for your comments and remarks. We hereby address the each points that was present the annotated pdf file. We also did an overall check for the english, organisation and mistyping.

**Detailed Answers to remarks and comments**

**Comment 1: "Do not indent paragraphs":**
Done accordingly through the paper.

**Comment 2: "lines 16-18 should be placed at the end of the Abstract, where general implications should be also mentioned".**
Modified accordingly, with small modifications "Integrating both geochronology and morphometrical results into lithospheric-scale numerical models allows a better understanding of this intraplate-orogen evolution and dynamic. We assume that the main conclusions are true to the general case of intraplate deformation. That is to say, once the topography has been generated by a triggering process. Associated uplift is then enhanced by erosion and isostatic adjustment leading to a significant accumulation of mainly vertical deformation.".

**Comment 3: "lines 19-22 to be placed after line 16"**
Changed accordingly.

**Comment 4, line 31: "of what"**
Simplification done with "part" changed with "unexplained uplift".

**Comment 5, line 31: "NOTE: dynamic topography refers to topography generated by movement caused by differentiaf buoyancy (convection) in the Earth's mantle."**
Agree. We argue that this dynamic topography could be one satisfactory explanation for the uplift triggering phenomenon, however our study has no quantification to assert this hypothesis.

**Comment 6: line 42: "Why focusing on the study area? Linkage between the previous lines is needed"**
We added theses sentences in the text for clarity: "Intraplate deformations evidenced by seismic activity is sometimes explained by the transient phenomenon (e.g., glacial isostatic rebound, hydrological loading). However, to explain the persistence through time of intraplate deformation, and explain the high finite deformation we can observe in the topography in many parts of the world as for instance the Ural mountains in Russia, the Blue Mountains in Australia or the French Massif Cen-tral, one needs to invoke continuous processes at the geological time-scale. Located in the southwestern Eura-sian plate (fig. 1), the French Massif Central is an ideal case to study this processes because a high resolution DEM encompasses the whole region and widespread karstic areas are present along its southern and western edges, allowing the possibility to quantify landscape evolution rates thanks to TCN burial ages.".

**Comment 7: line 58: "From line 58, this text should be included in a paragraph entiled "Geologial Backgorund". Reference to a regional figure is needed"**
The sub-section has been created. Link toward geological information website (from the french geological survey (BRGM) is mentioned.

**Comment 8: line 65 "age? Regional significance?", in reference to the Cevennes Fault system.**
"The area is also affected by the major NE-SW trending Cevennes fault system, a lithospheric-scale fault, inherit-ed from the Variscan orogen. This fault system was reactivated several times (e.g. as a strike-slip fault during the Pyrenean orogen or as a normal fault during the Oligocene extension)." was added for concise information.

**Comment 9: Suggested to delete lines 100 to 108**
Changed as suggested

**Comment 10: for line 265-267: "The first part should be put in the Materials and Methods section".**
This section can be, indeed redundant with previous section. However, because of the complex and multidisciplinary approach, we think that a short reminder of the inquired hypothesis helps the reader when starting this section.

**Comment 11: "The first part should be put in the Materials and Methods section" concerning the first lines of the section "Geomorphometrical approach".**
We slightly changed the organisation of the section in order to 1) increase the clarity and 2) highlight the multi-disciplinary approach. The point being that both section "Determining incision rate" and "Geomorphometry signature"

are stand-alone sections. This choice is also motivated by the fact that given the size of the paper, such organization should result in easier reading. See the modified document for changes.

**Comment 12: for conclusion reorganization "Present the main conclusions as bullet points" and "Which the main implications at broader scale: i.e. for intraplate deformation?".**
We modified the conclusion accordingly, see modified document.

Line 11: "dynamic" changed for "dynamics"

Line 12-13: space added between number and units

Line 13: question mark linked with "(e.g. Geodesy and Seismology)". Changed to geodetic and seismologic data.

Line 14 – 16: "Because the Cévennes margin allows the use of karst sediments geochronology and morphometrical analysis, we study the vertical displacements of that region: the southern part of the French Massif-Central" changed for "We focus our study on the southern part of the Massif-Central, known as the Cévennes and Grands Causses, which is a key area to study the relationship between the recent geological deformation and landscape evolution. This can be done through the study of numerous karst systems with trapped sediments combined with the anal-ysis of a high-resolution DEM."

Line 17: "helped" changed with "integrated" with

Line 30: "dating technics" changed as suggested by "the geochronological results" and "obtain" changed as suggested by "derived"

Line 31-32: "due to the Massif Central Plio-Quaternary magmatism" changed with "as consequence of the Pliocene-Quaternary magmatism of the region" as suggested

Line 32: "Plio-Quaternary" suggested to be changed by "Pliocene-Quaternary". Changed for all the occurrences in the manuscript

Line 35: "Such" changed as suggested by "this"

Line 36: "yrs." Changed with "yr"

Lines 37-41: "On geological time-scales, transient phenomenon that are classically used to explain intraplate deformations (as seen through the seismic activity) can not be a satisfactory explanation though, this then raises the question of the origin of the high finite deformations observed in many parts of the world as for instance the Ural mountains in Russia, the Blue Mountains in Australia or the French Massif Central." changed with "Intraplate deformations evidenced by seismic activity is sometimes explained by the transient phenomenon (e.g., glacial isostatic rebound, hydrological loading). However, to explain the persistence through time of intraplate deformation, and explain the high finite deformation we can observe in the topography in many parts of the world as for instance the Ural mountains in Russia, the Blue Mountains in Australia or the French Massif Central, one needs to invoke continuous processes at the geological time-scale.".

Line 52: "mountains" is removed.

Line 53-54: "(Topographic font in figure 1 show first order topography and morphology)" has been removed.

Line 54: "figure 1" changed with "Figure 1" as suggested.

Line 58: "formations" and "area" changed as suggested by "rock units" and "in the study area" respectively.

Line 60: "lower" is highlighted. Changed to middle.

Line 61: "thick" changed with "of thickness" as suggested

Line 63: "as being at the origin of" changed as suggested with "for the origin".

Line 65: "between" added as suggested.

Line 68: "Eventually" changed with "Finally".

Line 73 and 74: Removed as suggested.

Line 78: References moved to the end of the sentence as suggested.

Line 82: "in debate" changed as suggested by "debated"

Line 83: One reference, with overview synthesis was added.

Line 90: subsection title changed to "Materials and methods" as suggested

Line 93: "of" added as suggested

Line 93 to 95: "We employ two methods, cosmogenic 10Be/26Al burial dating quartz cobbles that have been transported by rivers and paleomagnetic analyses along vertical profiles of endokarstic clay both of which have been deposited in multiple cave systems at the time cave entry was at river channel elevation" rephrased to "We employ two methods to infer allochthonous karstic infilling age and associated river down-cutting. First, we use quartz cobbles to measure concentration of cosmogenic 10Be and 26Al isotopes. The 10Be/26Al ratio provide burial ages of these karstic infilling. Second, paleomagnetic analyses of clay deposits provide paleo-polarities. In both cases, vertical profiles among tiered caves systems and horizontal galleries could provide local incision rate information."

Line 95: "In parallel, by analyzing a high-resolution DEM (5m), we show that the region is affected by a regional tilting." modified to "By analyzing a high-resolution DEM (5m), we show that the region is affected by a southeastward regional tilting".

Line 108-109: Modified as suggested to "Our research approach provides an opportunity to discriminate between three possible explanations for the current terrain morphology."

**Line 109: "terrain morphology" proposed to be changed by morphotectonic signature.**
We choose to stay with "terrain morphology" since part of it could be only related to to climatic fluctuations, without a tectonic control. Morphotectonic terms would be misleading, providing a strong *a priori* interpretation.

Line 199: Subsection title changed to "Local incision rate from burial ages (Rieutord Canyon)"

Line 215: Subsection title changed to "Local incision rate from paleomagnetic data (Southern Grands Causses)"

Line 264: subsection title changed to "Geomorphometrical signature".
We consider that the term Geomorphometrical is more suited because it deals with morphological quantification

Line 268 "differential vertical movement" added for better clarity.

Line 269: "Fig. 9" is highlighted with "Fig. 9" as a comment. We are not sure of what we should do so we keep it as it is.

Line 271: "Other issue could be due to diffusion processes that could create apparent tilting." changed to "We point out that surface slope increase through time (e.g. apparent tilting) could be due to diffusion processes and not related to differential vertical displacements".

Line 274: "us" changed with use.

Line 312: Subsection "Numerical modelling" incorporated inside the "Discussion" subsection.

[revised manuscript text omitted]

Superimposed at the inheritance from Durancian event, the last two major tectonic episodes which are the Pyrenean compression and the Oligocene extensionshaped the large-scale structural morphology of the region. Afterwards during the Pliocene-Quaternary period, only intense volcanic activity has affected the region, from the Massif Central to the Mediterranean shoreline. This activity is characterizsed by several volcanic events that are well constrained in age (Dautria et al., 2010). The last eruption occurred in the Chaîne des Puys during the Holocene (i.e. the past 10 kyrs (Nehlig et al.,

2003; Miallier et al., 2004). Some authors proposed that this activity is related to a hotspot underneath the Massif Central  leading to an observed positive heat-flow anomaly and a possible regional plio cene-Quaternary uplift (Granet et al., 1995; Baruol and Granet, 2002). Geological mapping at different scale can be found at: http://infoterre.brgm.fr/.

——— Despite this well described overall geological evolution the onset of active incision that has shaped the deep valleys and canyons (e. g. Tarn or Vis river, Fig 1) across the plateaus, and the mechanisms that controlled this incision are still  debate d. One hypothesis proposes that canyon formation was driven by the Messinian salinity crisis with a drop of more than 1000m in

Mediterranean Sea level (Moccochain., 2007). This, however, would then not explain the fact that the

Atlantic watersheds show similar incision. Other studies suggested that the incision is controlled by the collapse of cave galleries that lead to fast canyon formation mostly during the late Quaternary, thus placing the onset of canyon formation only a few hundreds of thousands of years ago (Corbel,

1954). In contrast, it has also been proposed more recently (based on relative dating techniques and sedimentary evidence) that incision during the Quaternary was negligible (i.e. less than a few tens of meters), and that the regional morphological structures seen today occurred around 10 Ma (Séranne et al., 2002; Camus, 2003).

1.3 Materials and methods

——— In this paper, we provide new quantitative constraints on both the timing of incision and the rate of river down-cutting in the central part of the Cévennes and of the Grands Causses that has resulted in the large relief between plateau and channel bed. -

We employ two methods,  to infer allochthonous karstic infilling age and associated river down- cutting. First, we use quartz cobbles to measure concentration of cosmogenic [10]Be and  [26]Al isotopes. The [10]Be/[26]Al ratio provide burial ages of these karstic infilling. Second, squartz cobble  paleomagnetic analyses of clay deposits provide paleo-polarities. In both cases, vertical profiles among tiers caves systems and horizontal galleries could provide local incision rate information.

In parallel, y analyzing a high- resolution DEM (5m), we show that the region is affected by a southeastward  regional tilting. Our results allow to quantify the role of the Plio cene-Quaternary incision on the Cévennes landscape evolution and to constrain numerical modeling from which we derive the regional uplift rates and a tilt of geomorphological markers.

If incision is initiated by uplift centeredcentred on the North of the area where elevations are
maximum, it will lead to tilting of fossilizedfossilised topographic markers as strath terraces. Our
research approachmethod of analyses provides an opportunity to selectdiscriminate 
[revised manuscript text omitted]